

**Direct measurement of $N_2O_5$ heterogeneous uptake coefficients on ambient**
**aerosols via an aerosol flow tube system: design, characterization and**
**performance**
Xiaorui Chen[1,a], Haichao Wang[3,4*], Tianyu Zhai[1], Chunmeng Li[1], Keding Lu[1,2*]
[1]State Key Joint Laboratory of Environmental Simulation and Pollution Control, College of
Environmental Sciences and Engineering, Peking University, Beijing, China.
[2]The State Environmental Protection Key Laboratory of Atmospheric Ozone Pollution Control,
College of Environmental Sciences and Engineering, Peking University, Beijing, China
[3]School of Atmospheric Sciences, Sun Yat-sen University, Zhuhai, 519082, China
[4]Guangdong Provincial Observation and Research Station for Climate Environment and Air
Quality Change in the Pearl River Estuary, Key Laboratory of Tropical Atmosphere-Ocean
System, Ministry of Education, Southern Marine Science and Engineering Guangdong
Laboratory (Zhuhai), Zhuhai, 519082, China
[a]now at: Department of Civil and Environmental Engineering, The Hong Kong Polytechnic
University, Hong Kong, China
*Correspondence to:* Haichao Wang (wanghch27@mail.sysu.edu.cn), Keding Lu
(k.lu@pku.edu.cn)
**Abstract.** An aerosol flow tube system coupled with detailed box model was newly developed
to measure $N_2O_5$ heterogeneous uptake coefficients ($\gamma(N_2O_5)$) on ambient aerosols directly.
This system features simultaneous measurements of $N_2O_5$ concentration at the both entrance
and exit of the flow tube to ensure an accurate derivation of $N_2O_5$ loss in the flow tube.
Simulation and laboratory tests demonstrate that this flow tube system is able to overcome the
interference from side reactions led by varying reactants (e.g., $NO_2$, $O_3$ and $NO$) and improve
the robustness of results with the assistance of box model method. Factors related to $\gamma(N_2O_5)$
derivation were extensively characterized, including particle transmission efficiency, mean
residence time in the flow tube and wall loss coefficient of $N_2O_5$, for normal operating



condition. The measured $\gamma(N_2O_5)$ on $(NH_4)_2SO_4$ model aerosols were in good agreement with
literature values over a range of relative humidity (RH). The detection limit of $\gamma(N_2O_5)$ was
estimated to be 0.0016 at low aerosol surface concentration (Sa) condition of 200 $\mu m^2$ $cm^{-3}$.
Given the instrument uncertainties and potential fluctuation of air mass between successive
sampling modes, we estimate the overall uncertainty of $\gamma(N_2O_5)$ that ranges from 16 to 74%
for different ambient conditions. This flow tube system was then successfully deployed for
field observations at an urban site of Beijing influenced by anthropogenic emissions. The
performance in field observation demonstrates that the current setup of this system is capable
of obtaining robust $\gamma(N_2O_5)$ amid the switch of air mass.

## 37    1 Introduction

Dinitrogen pentoxide ($N_2O_5$), forming from the reaction of nitrogen dioxide ($NO_2$) and nitrate
radical ($NO_3$), acts as an important reservoir of atmospheric nitrogen. The $N_2O_5$ can undergo
either thermal dissociation (back to $NO_2$ and $NO_3$; photolysis of $NO_3$ also generate $NO_2$) to
release $NO_2$ or hydrolysis (both homogeneous and heterogeneous) to remove nitrogen oxides
from the atmosphere (Brown and Stutz, 2012;Chang et al., 2011). Among the budgets of $N_2O_5$,
the uptake on aerosol particles is a highly efficient pathway to be responsible for production
of nitrate aerosol in some regions (Fu et al., 2020;Wang et al., 2019;Wang et al.,
2017c;Baasandorj et al., 2017;McDuffie et al., 2019;Prabhakar et al., 2017;Wang et al.,
2018a;Chen et al., 2020) and promote activation of chlorine via $ClNO_2$ formation (Bertram
and Thornton, 2009a;Osthoff et al., 2008;Tham et al., 2018;Thornton et al., 2010;Wang et al.,
2017f). The $N_2O_5$ uptake coefficient ($\gamma(N_2O_5)$) is critical in determining the uptake reaction
rate of $N_2O_5$ on aerosol in addition to aerosol surface area (Sa). It represents the fraction of
collisions between gaseous $N_2O_5$ molecules and particle surfaces that resulted in a loss of $N_2O_5$.
Model simulation showed the variations in $\gamma(N_2O_5)$ can significantly influence the fate of NOx,
$O_3$ and OH radical in a regional (Li et al., 2016;Sarwar et al., 2012;Lowe et al., 2015) and
global scale (Dentener and Crutzen, 1993;Evans and Jacob, 2005;Macintyre and Evans,
2010;Murray et al., 2021). However, ambient data of direct observation on $\gamma(N_2O_5)$ is still



scarce. It is thereby necessary to develop an accurate equipment or method to quantify this
parameter on ambient aerosols.

Extensive laboratory experiments have been conducted to derive the values of $\gamma(N_2O_5)$ on

aerosols and understand the mechanism of $N_2O_5$ uptake by various methods, including aerosol
flow reactor (Kane et al., 2001;Mozurkewich and Calvert, 1988;Hu and Abbatt,
1997;Thornton and Abbatt, 2005;Thornton et al., 2003;Tang et al., 2014;Bertram and Thornton,
2009a), droplet train reactor (Van Doren et al., 1990;Schweitzer et al., 1998), Knudsen flow
reactor (Karagulian et al., 2006) and smog chamber (Wahner et al., 1998;Wu et al., 2020). The
$\gamma(N_2O_5)$ was found to be highly variable and dependent on particle chemical composition,
acidity, phase state and the presence of organic coating using these laboratory methods under
controllable conditions (Badger et al., 2006;Bertram et al., 2011;Fried et al., 1994;Griffiths et
al., 2009;Gross et al., 2009;Hallquist et al., 2000;McNeill et al., 2006;Mentel et al.,
1999;Riemer et al., 2003). While laboratory results have contributed to recognize the
mechanism of $N_2O_5$ uptake and develop $\gamma(N_2O_5)$ parameterizations (Anttila et al.,
2006;Bertram and Thornton, 2009b;Davis et al., 2008;Griffiths et al., 2009;Riemer et al.,
2009), issues might emerge when quantitatively extended to ambient conditions due to the
discrepancy between laboratory conditions and real air mass. For example, much higher
reactant and particle concentration usually used in laboratory experiments might induce
surface saturation or secondary reactions in a short time period, which lead to the bias of
reaction rate used in ambient conditions. In addition, the physicochemical properties of
ambient aerosol are much more complicated that the model aerosol used in laboratory studies,
which led to the laboratory results on model aerosols are difficult to accurately represent what
happens on the atmospheric aerosols.

There have been several methods implemented for field campaigns to indirectly derive

$\gamma(N_2O_5)$, simply based on observation of ambient $NO_3$, $N_2O_5$, $NO_2$, $O_3$, $ClNO_2$, $pNO_3^-$ and
other auxiliary parameters without special equipment to capture the decay of $N_2O_5$ like
laboratory ways. These include (1) the linear fit between $N_2O_5$ ($NO_3$) lifetime and the product
of $NO_2$ and Sa concentration according to steady state equations (Brown et al., 2002;Brown et



al., 2009;Brown et al., 2006;Platt et al., 1984;Wang et al., 2017b;Wang et al., 2017d;Tham et
al., 2016;Wang et al., 2017f;Brown et al., 2016), (2) the analysis of production rates of
products (pNO$_3^-$ and ClNO$_2$) resulting from N$_2$O$_5$ uptake under a stable condition (Mielke et
al., 2013;Phillips et al., 2016;Wang et al., 2018b) and (3) box model simulations with an
iterative approach to reproduce the evolutions of NO$_3$-N$_2$O$_5$ chemistry within each separate
air mass after sunset (McDuffie et al., 2018;Wagner et al., 2013;Wang et al., 2020a;Yun et al.,
2018). All these methods contain some specific assumptions and are only applicable in a few
special cases.
To directly determine the γ(N$_2$O$_5$) on ambient aerosols, Bertram et al. (2009a) firstly
design an entrained aerosol flow reactor to adapt for low atmospheric Sa concentration with
easy operation. By switching between filtered and bypass sampling mode, the N$_2$O$_5$
concentration at the exit of flow tube can be measured in the presence and absence of aerosols,
respectively. The pseudo-first-order rate coefficients for N$_2$O$_5$ loss on aerosols is thereby
derived from the ratio of measured N$_2$O$_5$ concentration in these two modes within a duty cycle
according to Eq. 1:

$$k_{aerosols} = -\frac{1}{\Delta t} \ln \frac{[N_2O_5]_{\Delta t}^{w/particles}}{[N_2O_5]_{\Delta t}^{wo/particles}} \qquad \text{Eq. 1}$$

where the $\Delta t$ is the mean residence time of the flow tube, and the $[N_2O_5]_{\Delta t}^{wo/particles}$ and
$[N_2O_5]_{\Delta t}^{w/particles}$ are the measured N$_2$O$_5$ concentration at the exit of flow tube in filtered and
bypass mode, respectively. Assuming the gas-phase diffusion effect is negligible for
atmospheric particles and low reaction probability (γ<0.1) (Fuchs and Sutugin, 1970), γ(N$_2$O$_5$)
can then be calculated from Eq. 2:

$$\gamma(N_2O_5) = \frac{4 \times k_{aerosols}}{c \times S_a} \qquad \text{Eq. 2}$$

This method was deployed to measure γ(N$_2$O$_5$) on ambient particles during two field
campaigns (Bertram et al., 2009b;Riedel et al., 2012) and on aerosols generated in the
laboratory (Ahern et al., 2018;Mitroo et al., 2019). While values of γ(N$_2$O$_5$) were determined



to be robust in laboratory experiments, most of data would be dropped under ambient
conditions due to the variations of wall loss coefficients (dominated by RH), fresh NO
emission, N$_2$O$_5$ regeneration and flow pattern inside the flow tube. Based on the above
measurement system, Wang et al. (2018c) added NOx, O$_3$ and Sa measurement on the exit of
flow tube and introduce an iterative box model to minimize the potential influences from
changing air mass and non-linear response of interference reactions. With the assumption of
the equilibrium between NO$_3$ and N$_2$O$_5$, the box model runs backward and forward iteratively
to obtain the N$_2$O$_5$ loss rate constant in the absence ($k_{het}^{wo/particles}$) and presence ($k_{het}^{w/particles}$)
of aerosols respectively. The difference between these two parameters can finally derived the
γ(N$_2$O$_5$) according to Eq. 3, assuming the wall loss effect stays consistent.

$$\gamma(N_2O_5) = \frac{4(k_{het}^{w/particles} - k_{het}^{wo/particles})}{c \times S_a} \qquad \text{Eq. 3}$$

This iterative approach was demonstrated to be able to buffer against certain fluctuations of
air mass and measure γ(N$_2$O$_5$) in the polluted atmosphere (Yu et al., 2020b).
Until now, only few direct measurements of γ(N$_2$O$_5$) on ambient aerosols have been
conducted during field campaigns (Bertram et al., 2009b;Riedel et al., 2012;Yu et al., 2020a).
Even though combining with dataset from indirect approaches (e.g. steady state
approximations), it is still challenging to characterize the temporal and spatial distributions of
γ(N$_2$O$_5$) on ambient aerosols. To better investigate the reactive uptake of N$_2$O$_5$ on aerosols in
different environments, we develop an aerosol flow tube system with newly designed gas
circuit and data acquisition procedures to quantify γ(N$_2$O$_5$) on ambient aerosols. In the
following sections, the setup of this system and laboratory characterizations for each part are
described in details. Procedures of acquiring and processing data are compared to previous
methods and discussed with potential uncertainties. Laboratory tests on model aerosols and
field observations are presented to demonstrate its performance under varying ambient
conditions.


## 2 The aerosol flow tube system

A schematic of the aerosol flow tube system is shown in Figure 1. The ambient air enters
the system from the sampling manifold, mixes with gaseous $N_2O_5$ source in a Y-tee and flows
to aerosol flow tube and detection instruments, as indicated by arrows in the figure. The system
is distinct from previous flow tube systems due to its continuous monitor of NOx and $O_3$
concentration before the inlet of flow tube (after sampling air mixing with $N_2O_5$ source) and
the simultaneous measurements of $N_2O_5$ concentration both at the inlet and the exit of flow
tube within a duty cycle. Constraints of these variables during the subsequent data processing
can enhance the measuring accuracy.

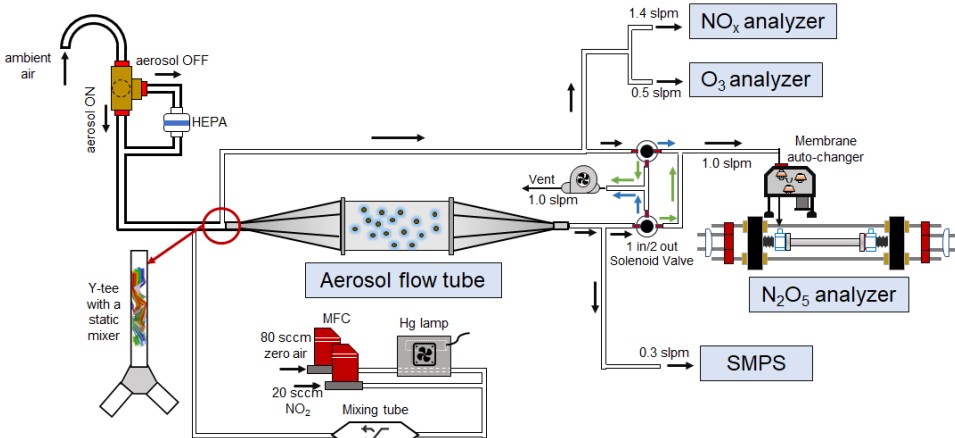


**Figure 1.** Overall schematic of aerosol flow tube system. The arrows alongside the tube show
the flow directions. The black arrows indicate the flow directions consistent during the
measurements, green arrows indicate the flow directions active in measuring the exit $N_2O_5$
and blue arrows indicate the flow directions active in measuring the inlet $N_2O_5$.

## 2.1 Sampling manifold

The sampling tube is made of a 50 cm long and half inch outside diameter (OD) aluminum
tubing, with a curve tip (10 cm radius of curvature) turning the inlet straight down in order to
avoid precipitation. The ambient air is then pass through a three-way solenoid ball valve,
which is controlled by a time relay to either allow the air to flow directly into a following Y-



tee (filter bypass mode) or divert to a HEPA (high efficiency particulate air filter, Whatman)
to remove particles (filter inline mode). We choose a stainless-steel ball valve with the same
OD as the sampling tube to minimize the particle loss in filter bypass mode. The HEPA can
retain particles at a high efficiency (>99.9%) with low pressure drop and RH difference
between filter inline and bypass mode.

## 154   **2.2 Gaseous $N_2O_5$ generation**

A home-made temperature-controlled gas generator is used to generate gaseous $N_2O_5$ in-situ
via the reaction of $O_3$ with $NO_2$ (R1) and the subsequent reaction of produced $NO_3$ with $NO_2$
(R2).

$$NO_2 + O_3 \rightarrow NO_3 + O_2 \qquad\qquad\qquad\qquad \text{R1}$$

$$NO_2 + NO_3 + M \leftrightarrow N_2O_5 + M \qquad\qquad\qquad\qquad \text{R2}$$

$NO_2$ is delivered from a compressed gas cylinder (20 ppmv in $N_2$ diluent gas, Jinghao Corp.).
$O_3$ is generated from the photolysis of $O_2$ in compressed ultra-pure synthetic zero air at 254
nm, using a commercial mercury lamp (UVP, the USA) fixed inside the generator. The
produced $O_3$ are then mixed with $NO_2$ in a small darkened Teflon reaction tube for about 2
min under the temperature of 15 °C, stabilized by a Peltier cooler controlled by a proportion
integration differentiation algorithm. A PFA tube with polyethylene foam was used to transmit
the synthesized $N_2O_5$ to sampling stream and minimize the influence of ambient temperature
variation on $N_2O_5$ level. The flow rate of $NO_2$ (20 sccm) and zero air (80 sccm) are controlled
by mass flow controller separately at a total of 100 sccm. By changing the flow rate ratio
between $NO_2$ and zero air, the generator can produce $N_2O_5$ concentration varying from 1 ppbv
to 6 ppbv (after dilution in zero air at sampling flow rate of 4.5 slpm). Under the typical
measurement condition, an excess of $NO_2$ concentration is applied to shift the equilibrium
towards $N_2O_5$ production (R2) and suppress the $NO_3$ concentration to less than 30 pptv, which
is expected to decrease the uncertainty of varying $NO_3$ reactivity (NO, VOCs and
heterogeneous loss). The resulted initial $N_2O_5$ concentration was 4.0 ppbv at the inlet of
aerosol flow tube, together with around 50 ppbv of $NO_2$ and 15 ppbv of $O_3$. A stability test on





$N_2O_5$ source showed the variation was within 1% for a 24-h continuous operation, with
ambient temperature ranging from 0 to 15 ℃.

**2.3 Aerosol flow tube**

Air flow enters and exits the flow tube via two identical conical diffuser caps at a diffuser
angle of 45°. A 35cm×14 cm inner diameter (ID) cylindrical tube is mounted in the middle
of two caps, flanged with screws and nitrile rubber O-rings. All sections of this aerosol flow
tube are made of stainless-steel with electro-polished and FEP-coated inside. The exterior of
the flow tube is insulated with aluminum coated polyethylene foam 3 cm thick to minimize
thermal eddies fluctuation of ambient temperature. Under the typical flow rate of 2.1 SLPM
in the flow tube, the axial velocity in the cylindrical tube section is 0.23 cm·s$^{-1}$ which produces
a Reynolds numbers ($Re$) of 22, well below the threshold of laminar flow ($Re$<2100).

In front of the flow tube, the synthesized $N_2O_5$ source is introduced perpendicular to

ambient air sampling stream via a regular stainless-steel tee and then the mixture enters a
stainless-steel Y-tee for further mixing. The inner surface of both regular tee and Y-tee is
electro-polished and coated with SilcoNert 2000 (Silotek Corp.), a technique commonly
applied in semiconductor industry, to maintain the transmission efficiency of particles and
minimize the loss of $N_2O_5$ in the meantime. A 10 cm long stainless-steel static mixer is
mounted inside the Y-tee in order to swirl the flow and thus facilitate the mixing between
sampling stream and $N_2O_5$ source in a relatively short distance. The presence of static mixer
at the inlet also help to improve the flow expansion performance after entering the flow tube
by minimizing flow recirculating towards the wall, which decreases the wall loss of $N_2O_5$ and
particles (Huang et al., 2017). After passing through the static mixer, the mixture of ambient
air and $N_2O_5$ source is split into two flows at the same flow rate, one of which straightly enters
the aerosol flow tube and the other one is diverted to measurements of NOx, $O_3$ and $N_2O_5$. We
measured the concentrations of NOx, $O_3$, $N_2O_5$ and Sa at the both exits of Y-tee under typical
flow rate for three repeated experiments (Figure 2). Almost the same gaseous concentrations
and particle distributions at both exits of Y-tee demonstrate that the $N_2O_5$ source has been well
mixed with the sampling flow and species concentrations at the inlet of flow tube can be





accurately determined via the measurements at the other exit of Y-tee.

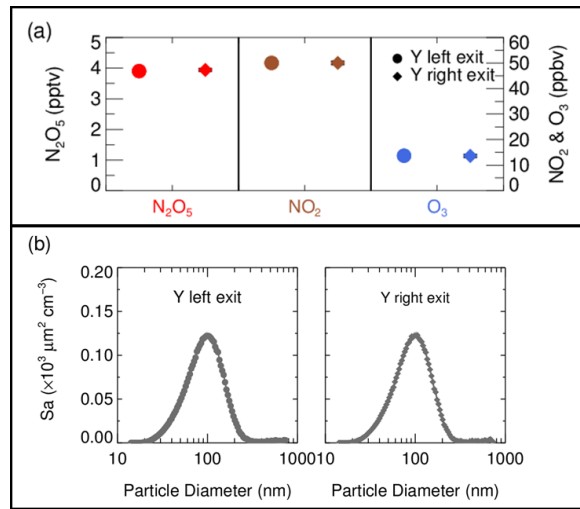


**Figure 2.** (a) The concentration of $N_2O_5$, $NO_2$ and $O_3$ in the mixture of $N_2O_5$ source and
sampling aerosols measured at each exit of Y-tee; (b) The size distribution of Sa concentration
in the mixture of $N_2O_5$ gas source and sampling aerosols measured at each exit of Y-tee.

## 2.4 Detection instruments

Instruments used in this system are listed in Table 1. A portable cavity-enhanced absorption
spectrometer (CEAS) is used to measure $N_2O_5$ concentration (Wang et al., 2017a) at both inlet
and exit of the aerosol flow tube by automatically switching the flow directions (see details in
section 2.5). Briefly, the $N_2O_5$ is thermally decomposed to $NO_3$ by heating up to 130°C and
then quantified according to the extinction coefficient caused by $NO_3$ absorption in the
wavelength window from 640 to 680 nm. A Teflon polytetrafluoroethylene (PTFE) membrane
is placed in front of the CEAS to remove particles, which will be replaced with a new one
every two hours by a self-designed membrane auto-changer. Laboratory tests have been
conducted to quantified the transmission efficiency of $N_2O_5$ over the membrane (92±3%),
sampling tube of CEAS (99.7%) and the inside of CEAS (93.6%). The detection limit of $N_2O_5$
was determined to be 2.7 pptv (1σ, 60s) and the measurement uncertainty was 19%. A time-
resolution of 60 s for $N_2O_5$ data acquisition is typically used to derive γ($N_2O_5$) in this study.
The CEAS has been successfully applied to measure ambient $N_2O_5$ concentration in several





field campaigns and laboratory studies (Chen et al., 2020;Wang et al., 2020a;Wang et al.,
2017b;Wang et al., 2020b;Wang et al., 2018b;Wang et al., 2022).
**Table 1.** Performance of related instruments incorporated in the flow tube system.

| Parameter | Technique | Time resolution | Detection Limit($1\sigma$) | Accuracy |
|---|---|---|---|---|
| NO | Chemiluminescence[a] | 1 min | 200 pptv | ±10% |
| $NO_2$ | Chemiluminescence | 1 min | 300 pptv | ±10% |
| $O_3$ | UV photometry | 1 min | 500 pptv | ±5% |
| VOCs | GC-MS/FID[b] | 60 min | 20-300 pptv | ±15% |
| $N_2O_5$ | CEAS | 1 min | 2.7 pptv | ±19% |
| Sa | SMPS | 5 min | - | ±10% |
| RH&T | Sensor | 1 min | - | ±0.1%&±0.1K |

[a] Photolytic conversion to NO through blue light before detection; [b] Gas chromatography
equipped with a mass spectrometer and a flame ionization detector;

At the inlet of flow tube, NOx concentration is measured via chemiluminescence method

equipped with a blue-light photolytic converter (Thermo, Model 42i) and $O_3$ concentration is
also measured via chemiluminescence method by adding excessive NO (Teledyne API, Model
T265). Both NOx and $O_3$ concentration are averaged to 1 min time-resolution. The size
distribution of particle number density is measured at the exit of flow tube using a scanning
mobility particle sizer (SMPS, TSI 3776), which determines the total Sa concentration
covering the range from 13 to 730 nm. Particles larger than this range usually contributed less
than 5% of total Sa according to our previous field measurements (Chen et al., 2020) and it is
included in the uncertainty analysis (see section 5). A cycle of size scanning is set to around 5
min and the derived Sa concentration is then interpolated into 1 min for further calculation.
Aerosols pass through a Nafion tubing (MD-700) before entering into SMPS to reduce RH to
less than 30%. The dry-state Sa is therefore corrected to wet-state at the RH inside the flow
tube for particle hygroscopicity. The growth factor, $f(\text{RH})=1+8.77\times(\text{RH}/100)^{9.74}$, used for
correction is valid only when RH is within the range from 30 to 90% (Liu et al., 2013). The
RH and temperature of flow are continuously measured both before entering and after leaving
the flow tube by commercial sensors (Rotronic, Model HC2A-S). The averages of the values



obtained at both locations are used to represent the RH and temperature inside the flow tube.
In addition, ambient volatile organic compounds (VOCs) are measured in-situ alongside the
aerosol flow tube system using an online gas chromatograph mass spectrometer coupled with
a flame ionization detector (GCMS-FID) to derive the $NO_3$ reactivity to VOCs ($k_{NO3-VOCs}$) in
the flow tube.

## 2.5 Procedures of data acquisition

The $N_2O_5$ concentration is acquired at both inlet and exit of the flow tube within a duty cycle
via a CEAS instrument, which is different from that only at the exit of the flow tube in previous
studies (Bertram et al., 2009a;Wang et al., 2018c). At each duty cycle, consisting of once HEPA
inline mode for measuring $k_{wall}$ of $N_2O_5$ and once HEPA bypass mode for retrieving the $N_2O_5$
loss on aerosols, the procedure that measuring $N_2O_5$ at the inlet of flow tube followed by that
at the exit is executed twice with one for each mode. An exemplary case obtained during a
field campaign is shown in Figure 3 to explain this procedure. Within the mode of HEPA inline,
$N_2O_5$ data is firstly acquired at the inlet of the flow tube and then switch to the exit of the flow
tube. The $k_{het}^{wo/particles}$, which is the $k_{wall}$ of $N_2O_5$, can be therefore derived from a box model
constrained by these $N_2O_5$ data (see section 3 for the model description and data processing).
The same procedures are executed in the mode of HEPA bypass, except the $\gamma(N_2O_5)$ is derived
according to Eq 2. Two three-way valves controlled by a time relay were implemented to
realize this procedure in order to avoid the changes of flow condition in the flow tube that
could have been caused. As indicated in Figure 1, the blue arrows show the flow directions
when measuring the $N_2O_5$ concentration at the inlet of flow tube, while the green arrows shows
that for the exit of flow tube. It should be noted that the concentration of NOx and $O_3$ are
always acquired at the inlet of the flow tube and the Sa concentration always at the exit of the
flow tube during the operation.

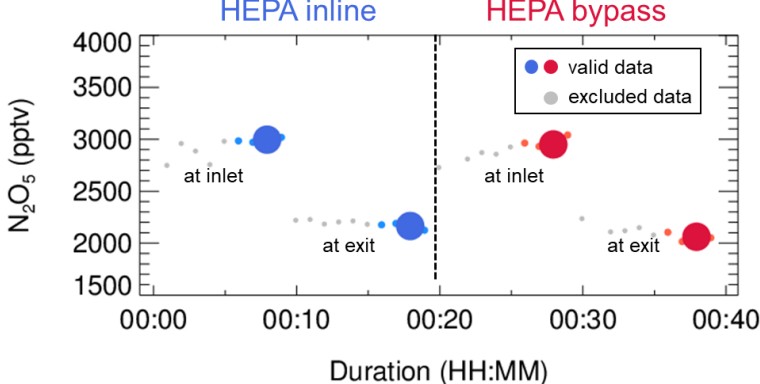


**Figure 3.** An exemplary case of measured $N_2O_5$ concentration within a duty cycle. This case

was observed on the night of 13 December 2020, with average ambient Sa of 320 $\mu m^2\ cm^{-3}$.

The derived $k_{wall}$ of $N_2O_5$ and $\gamma(N_2O_5)$ were 0.0023 $s^{-1}$ and 0.035, respectively. The blue dots

indicate $N_2O_5$ concentration measured under the mode of HEPA inline either at the inlet or

exit of the flow tube (denoted as texts); the respective averages (blue dots of larger size) are

used for deriving $k_{wall}$ (blue square). The red dots indicate $N_2O_5$ concentration measured under

the mode of HEPA bypass either at the inlet or exit of the flow tube; the respective averages

(red dots of larger size) are used for deriving the overall rate constant of $N_2O_5$ loss on the wall

and aerosols. The data points in gray are excluded from calculation due to unstable conditions

in the flow tube.

In addition, laboratory tests were conducted to determine a suited duration for each duty
cycle. During a duty cycle, the duration for each mode should last long enough to develop a
stable flow condition for particles or empty particles, while a much longer duration could
decrease the measurement time-resolution and leads to large uncertainty due to the fluctuations
within a long time period. We measured Sa and $N_2O_5$ concentration continuously at the exit of
flow tube when sampling $(NH_4)_2SO_4$ aerosols. As shown in Figure 4, it took about 15 minutes
for particles to rise to a stable level from none or to decrease from a certain level to none, when
our system underwent mode switches. The correspondingly periodical variation of $N_2O_5$
concentration was consistent with particles. The residence time distribution (RTD) profiles
(see in section 4.2) also demonstrated that it required at least 10 minutes for gaseous species
to evolve to a stable condition after a mode switch. As a result, a typical duration of duty cycle
is determined to be 40 minutes with 20 minutes for each mode. The $N_2O_5$ measurement at the
exit of the flow tube in the last 5 minutes of each mode is able to represent valid decays of
$N_2O_5$ under this mode and satisfy the requirements of further data processing.

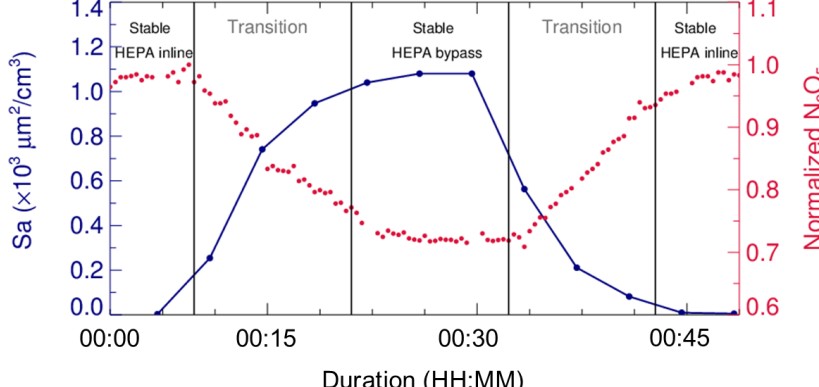


**Figure 4.** Variations of Sa and $N_2O_5$ concentration (normalized to peak values) measured at
the exit of flow tube when switching the sampling mode. The phases of species concentrations
in the flow tube approaching stable after a mode switch are denoted as the transition phases.
**3 Box model for determination of loss rate coefficients of $N_2O_5$**
**3.1 Method description**
Large uncertainties were found in retrieving $\gamma(N_2O_5)$ on ambient particles according to Eq. 1
in a previous flow tube study (Bertram et al., 2009a), due to the dependence of homogeneous
reaction rates on sampling modes and the atmospheric variations of parameters related to $NO_3$-
$N_2O_5$ chemistry (e.g. NO, $NO_2$, $O_3$, VOCs, and RH). To minimize these influences, a time-
dependent box model constrained by the measurements of $N_2O_5$ concentration and other
auxiliary parameters is applied to calculate loss rate coefficients of $N_2O_5$ under the mode of
HEPA inline and bypass, respectively. The model is able to simulate the reactions related to
budgets of $NO_3$-$N_2O_5$ chemistry in a dark condition, including R1, R2 and the follows:

$$NO_3 + NO \rightarrow 2\ NO_2 \hspace{4cm} R3$$

$$NO_3 + VOCs \rightarrow products \hspace{3cm} R4$$



$$N_2O_5 + \text{aerosols or wall} \rightarrow \text{products} \qquad\qquad R5$$

The rate constants for reactions R1 to R3 are referenced to IUPAC database. The reaction of
VOCs and $NO_3$ is treated as pseudo-first-order with a rate constant of $k_{NO3\text{-}VOCs}$, which is
calculated from the sum of rate constants for reactions of $NO_3$ with each VOCs scaled by the
concentration of VOCs measured by GC-FID. Due to low time-resolution of VOCs
measurements (1 h), the $k_{NO3\text{-}VOCs}$ is kept constant for each derivation of $\gamma(N_2O_5)$. The
suppressed $NO_3$ concentration is expected to attenuate the influence resulted from the
uncertainty of $k_{NO3\text{-}VOCs}$ (see discussion in section 5). The reaction R5 represents the loss of
$N_2O_5$ only on the wall in the mode of HEPA inline or on the both wall and particles in the
mode of HEPA bypass. The rate constant of R5 is also treated as pseudo-first-order and it is
adjustable among different runs.
The same procedures of data screening and model operation are applied to both sampling
and bypass modes, as shown in Figure 5. For example, in the mode of HEPA inline, the average
of NO concentration less than 6 ppbv and the variation of $N_2O_5$ measured at the inlet of flow
tube less than 10% should be validated prior to the following model operation. Under typical
concentration of $N_2O_5$ source we used in this flow tube system, the exit concentration of $N_2O_5$
is detected to be under triple detection limit with initial NO large than 6 ppbv according to our
laboratory tests. In ambient condition, high level of NO is usually also accompanied by rapid
variation due to fresh emission, which disturbs the decay of $N_2O_5$ in the flow tube and leads
to large uncertainty in deriving its loss rate coefficient. Excluding the cases that $N_2O_5$
measured at the inlet of flow tube varies exceeding 10% can further minimize the uncertainty
of $N_2O_5$ loss rate coefficient resulted from rapid change of $NO_3$ reactants (NO, VOCs). If the
measured data within the duration of a sampling mode satisfies the criteria for data screening
described above, the model can therefore simulate the reactions starting from the entrance of
flow tube and lasting for 156 s (mean residence time) based on these data. The initial
concentrations of $[NO]_{t=0}$, $[NO_2]_{t=0}$, $[O_3]_{t=0}$ and $[N_2O_5]_{t=0}$ are the averages of last-5-min values
measured at the inlet of flow tube. The RH and temperature are constrained by the mean values
during this sampling mode. By tuning the loss rate coefficient of $N_2O_5$ ($k_{N2O5}$) in the way of



binary search, we optimized an appropriate $k_{N2O5}$ to ensure that the $N_2O_5$ concentration output
from the simulation is consistent with last-5-min average of $N_2O_5$ concentration measured at
the exit of flow tube within 1 pptv. As a result, this derived $k_{N2O5}$ (aka. $k_{het}^{wo/particles}$) is
expected to be the $k_{wall}$ of $N_2O_5$. The same procedures above are then applied to the data
obtained in the mode of HEPA bypass, except that the derived $k_{N2O5}$ (aka. $k_{het}^{w/particles}$)
contains the loss rate coefficients of $N_2O_5$ on the both wall and particles. It should be noted
that the above calculation for obtained data is only valid under the variation of RH less than
2% within a duty cycle and the $k_{wall}$ of $N_2O_5$ can then be reasonably assumed to be constant
between two successive sampling modes. Therefore, the $\gamma(N_2O_5)$ can be retrieved by the Eq 3,
where the last-5-min averages of Sa concentration in the mode of HEPA bypass is used.

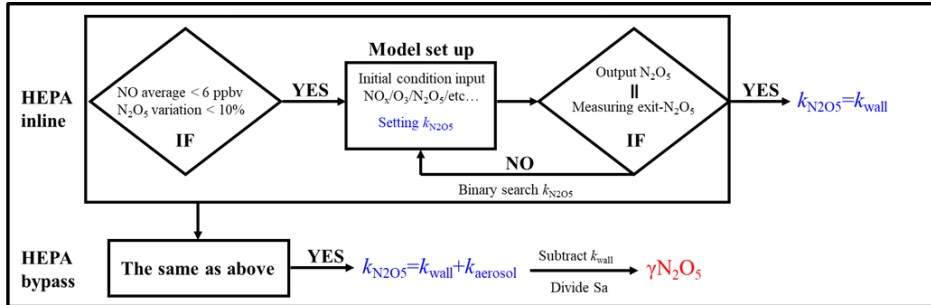


**Figure 5**. Flow diagram of $\gamma(N_2O_5)$ derivation through box model method.

## 344   3.2 Evaluation of the box model method

The box model method is introduced to our flow tube system in order to overcome the
influence from homogeneous reactions and variations of air mass on $\gamma(N_2O_5)$ derivation. A
series of scenarios were provided to evaluate the performance of box model method under
these influences. We allow NO, $NO_2$ and $O_3$ in the mixture of sampling air at the entrance of
the flow tube to vary in a reasonable range, in order to set up the scenarios of different gradients
of NO concentration and $NO_3$ production rates ($PNO_3$). Other relevant parameters were
prescribed to be fixed, including $N_2O_5$ source of 4 ppbv, Sa concentration of 1000 $\mu m^2 \cdot cm^{-3}$,
$\gamma(N_2O_5)$ of 0.02, $k_{wall}$ of 0.002 $s^{-1}$, $k_{NO3\text{-}VOCs}$ of 0.01 $s^{-1}$ and temperature 298K. By simulating
$N_2O_5$ evolutions in the flow tube based on these conditions, the exit concentration of $N_2O_5$



with and without particles will be obtained for various scenarios. The $\gamma(N_2O_5)$ on particles are
then calculated according to Eq 1&2 or by box model method used in this study. Comparison
between the results of two methods indicates that the box model method is more robust in
retrieving $\gamma(N_2O_5)$ due to the consideration of multiple reactions in the flow tube.

As shown in Figure 6(a), the exit concentration method ($\gamma(N_2O_5)$ exit-conc., derived

directly by Eqs. 1-2) underestimates $\gamma(N_2O_5)$ by varying degrees at different levels of $PNO_3$,
while the box model method can well reproduce the same $\gamma(N_2O_5)$ implemented in these
simulations (not shown). The $PNO_3$ is calculated from the initial $NO_2$ and $O_3$ concentration.
The degree of this underestimation for exit concentration method is mainly related to the in
situ $N_2O_5$ production in the flow tube. With a sustaining production of $NO_3$ via the reaction of
$NO_2$ to $O_3$ and rapid heterogeneous loss of $N_2O_5$ in the flow tube, the equilibrium between
$NO_3$ and $N_2O_5$ always shifts to the production of $N_2O_5$, and masking the actual amount of
$N_2O_5$ removal. In the mode of HEPA bypass, the $N_2O_5$ consumes faster than the other mode
due to the addition of particles, which further facilitates the $N_2O_5$ formation through the
equilibrium. Therefore, more $N_2O_5$ produced with particles in the flow tube leads to the
underestimation of $\gamma(N_2O_5)$ calculated by the exit concentration method. The increase of $PNO_3$
contributes to amplify this underestimation up to around 60% at a relatively polluted
environment, which could be encountered in southern China even during the wintertime.
Previous studies also found similar impacts from $N_2O_5$ production on retrieving $\gamma(N_2O_5)$ in
the aerosol flow tube (Bertram et al., 2009a;Wang et al., 2018c). In addition, the discrepancy
of $\gamma(N_2O_5)$ derived by two methods is much less dependent on the NO concentration at least
in the prescribed range of NO. As the ratio of $NO_3/N_2O_5$ is relatively small (<1%) in our $N_2O_5$
source, the difference of NO titration rates between two sampling modes has little impact on
$N_2O_5$ concentration and the subsequent $\gamma(N_2O_5)$ derivation.

To corroborate the results of comparison predicted by simulations, we also performed

laboratory tests to measure $\gamma(N_2O_5)$ on $(NH_4)_2SO_4$ aerosols using our flow tube system under
typical operating condition. The aerosols were conditioned to RH of 50% and doped with NO
gas (10 ppmv in $N_2$ diluent gas, Jinghao Corp.) at different gradients by diluted in ultrahigh-
purity $N_2$. Figure 6(b) shows that the $\gamma(N_2O_5)$ derived by box model method is at 0.01 and
consistent over the range of NO concentration from 0 to 6 ppbv applied to the tests. This result
agrees well with previous laboratory observation of $\gamma(N_2O_5)$ on $(NH_4)_2SO_4$ aerosols within
uncertainty (Badger et al., 2006;Hallquist et al., 2003;Kane et al., 2001). Similar to simulation
tests, the exit concentration method again underestimates $\gamma(N_2O_5)$ by 50 to 60% in the
laboratory tests, which reinforces that it is necessary for $\gamma(N_2O_5)$ measurements by an aerosol
flow tube to consider the interactions between homogeneous and heterogeneous reactions in
the flow tube using the box mode method especially with a long residence time. The absence
of dependence between NO concentration and $\gamma(N_2O_5)$ results also provides us the confidence
that this aerosol flow tube system is able to buffer against NO within the range from 0 to 6
ppbv under typical operating condition. However, this is not always the case when the NO is
accompanied with rapid fluctuations in a real atmosphere, which might lead to intractable
uncertainty and is therefore excluded from further analysis according to the criteria of data
screening described above.

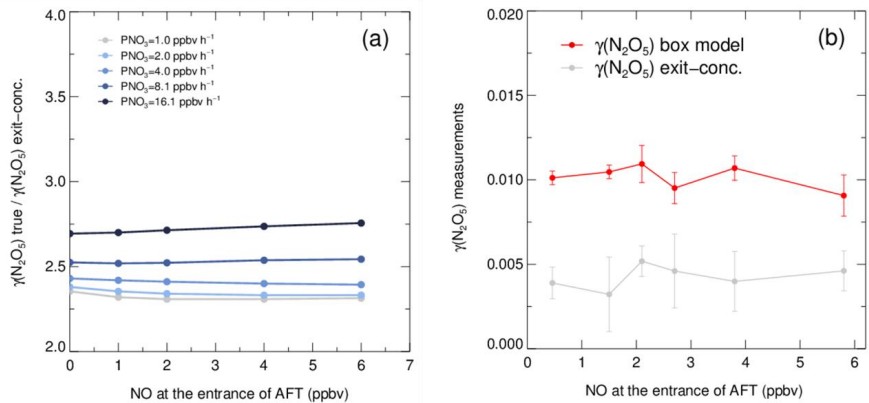


**Figure 6**. Simulated and laboratory tests on performance of box model method and exit
concentration method for $\gamma(N_2O_5)$ derivation. (a) The ratios of given $\gamma(N_2O_5)$ ($\gamma(N_2O_5)$ true)
over exit concentration derived $\gamma(N_2O_5)$ ($\gamma(N_2O_5)$ exit-conc.) determined from simulated
scenarios. The $\gamma(N_2O_5)$ derived by box model method is exactly the same as $\gamma(N_2O_5)$ true. The
ratios vary with NO concentration and the lines are color coded by $PNO_3$ values. Both NO
concentration and $PNO_3$ represent the values at the entrance of aerosol flow tube. (b) $\gamma(N_2O_5)$
measurements on lab-generated $(NH_4)_2SO_4$ aerosols under different gradients of NO with





constant RH of 50% and PNO3 typically generated from our $N_2O_5$ source. The red line shows
the $\gamma(N_2O_5)$ derived by box model method and gray line shows the $\gamma(N_2O_5)$ derived by exit
concentration method. The NO concentrations are measured at the entrance of aerosol flow
tube.
Overall, the introduction of box model method in this study is able to effectively avoid
the underestimation caused by the lack of consideration of side reactions in the flow tube.
Although an iterative box model, including backward and forward simulation, has been
successfully applied to realize the $\gamma(N_2O_5)$ measurements via an aerosol flow tube in polluted
environments (Wang et al., 2018c), the box model method combined with current flow tube
system in this study can improve measuring accuracy on some aspects. First, we simulate NO3-
$N_2O_5$ relationship via specific reactions rather than approximating it in equilibrium and
introducing the equilibrium coefficient ($K$eq) into calculation. Determining $NO_3$ or $N_2O_5$
concentration by $K$eq could induce large bias under the high aerosol loading and low
temperature (Chen et al., 2021). Second, it is more accurate to constrain the box model with
directly measured NOx, $O_3$ and $N_2O_5$ at the entrance of the flow tube. Under a real atmosphere,
the initial $N_2O_5$ concentration after mixing with sampling air is expected to be not as stable as
that observed in laboratory tests, due to the variations in temperature, NO concentration and
other related parameters. Numerical simulations based on a constant initial $N_2O_5$ and
estimation of initial concentrations for other species through backward simulation could then
lead to bias in the resulting $\gamma(N_2O_5)$ under these conditions.
**4 Laboratory characterizations**
**4.1 Particle transmission efficiency**
The transmission efficiency of particles in the sampling module and flow tube are estimated
respectively in Figure 7. In the laboratory, pure ammonia nitrate ($(NH_4)_2SO_4$) aerosols were
generated from an atomizer loading with 0.1 M $(NH_4)_2SO_4$ solution. The RH and concentration
of produced aerosols flow was conditioned in a glass bottle (~2 L) by introducing a humidified
dilution flow of ultrahigh-purity $N_2$. As a result, aerosols in different concentrations





(1000~4500 μm² cm⁻³) and under a range of RH (20~70%) were applied to test the
transmission efficiency. Figure 7(a) shows the loss of total Sa concentration in the sampling
module and flow tube are 8±1% and 10±2% on average, respectively. We found that the
fraction of particles loss is mainly caused by particles smaller than 100 nm. This is most likely
due to the turbulence generated by static mixer and the recirculation in the flow tube. Large
particles are prone to stay within the main flow direction, whereas small particles readily
adsorb on the walls by the entrainment of turbulence or recirculation. In addition, the particles
distribution measured at the exit of flow tube with HEPA inline (gray line in Figure 7(a))
demonstrated its capability of removing almost all particles (>99.5%) at the typical flow rate.
Transmission efficiency tests were also conducted on ambient aerosols (Figure 7(b)) and the
resulted loss of total Sa concentration was similar to that using laboratory-generated aerosols.

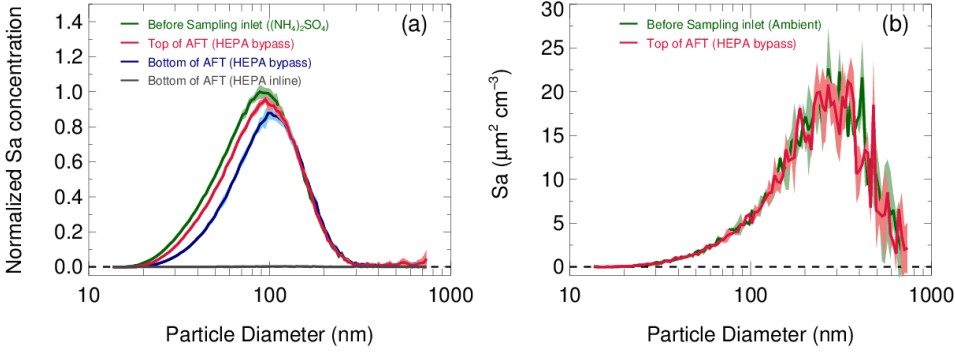


**Figure 7**. (a) Particles transmission determined by sampling laboratory-generated $(NH_4)_2SO_4$
aerosols. Aerosols at different concentrations and RH levels are used in experiments and the
size distribution of Sa concentration are normalized to the peak values. The normalized size
distribution of Sa concentration measured before sampling inlet (green line), at the inlet of
flow tube with HEPA bypass (red line) and at the bottom of flow tube with HEPA bypass
(blue line) are shown respectively. Under the mode of HEPA inline, the Sa concentration was
almost zero at the bottom of flow tube (gray line). The shadows indicate the standard
deviations of the normalized Sa concentration for all experiments. (b) Particles transmission
determined by sampling ambient particles.
**4.2 Residence time in the flow tube**
The method of residence time distribution (RTD) was applied to estimate the average reaction





time of the gas species in the flow tube (residence time). In comparison to ideal plug flow, the
RTD method can better describe actual behavior of the flow in practice and determine the
mean residence time more accurately (Danckwerts, 1953). Several studies have also used this
RTD method to determine the residence time in the flow tube (Huang et al., 2017;Wang et al.,
2018c;Lambe et al., 2011).

The RTD profiles were obtained by introducing a 2 s pulse of $NO_2$ gas diluted in $N_2$ into

the flow tube under RH less than 1%. $NO_2$ is relatively inert against the flow tube wall coated
with FEP and was measured at the exit of the flow tube by a CEAS (Li et al., 2021) at high
time-resolution (2 Hz). A three-way solenoid valve combined with a time relay was
implemented to control the pulse in order to avoid the disturbance on flow condition from the
injection. Experiments were performed under typical operation. The mean residence time ($t_{ave}$)
can be derived from the each RTD profile according to Eq. 4,

$$t_{ave} = \frac{\sum_{i=0} C_i \times t_i}{\sum_{i=0} C_i},$$
    Eq. 4

where the $C_i$ is the concentration of $NO_2$ recorded at the time step $t_i$. From the RTD profiles
of $NO_2$ injection experiments in Figure 8, the determined $t_{ave}$ was 156±3 s. This value is 19%
less than the space time ($\tau_{space}$, flow tube volume divided by operation flow rate, 192.6 s). It
has also been found that the assumption of ideal plug flow overestimated the residence time
in previous flow tube experiments (Lambe et al., 2011;Huang et al., 2017;Wang et al., 2018c),
which could lead to underestimation of the derived $k_{N2O5}$.

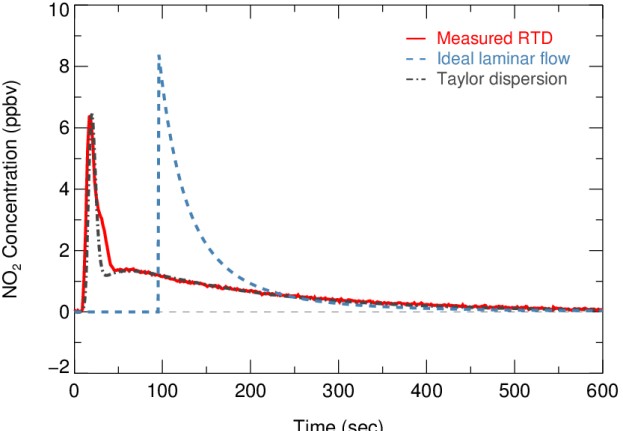


**Figure 8**. Residence time distribution derived by sampling $NO_2$ gas. Red solid line indicates the measured RTD profiles. The calculated RTD of ideal laminar flow (without dispersions) and the Taylor dispersion model fitted to measurements are shown as blue dash line and dot-dash line, respectively.

Two theoretical RTDs were calculated, namely ideal laminar flow and Taylor diffusion, besides the measured RTD, intending to reflect the fluid field inside the flow tube. The ideal laminar flow describes the flow without dispersion. The velocity profile of ideal laminar flow is parabolic, with the fluid in the center of the tube moving the fastest. According to the following Eq. 5, the RTD of ideal laminar flow is scaled by the integrated concentration of $NO_2$ and presented as the blue dash line in Figure 8.

$$\begin{cases} 0, & t < 0.5\tau_{space} \\ \dfrac{\tau_{space}{}^2}{2t^3}, & t \geq 0.5\tau_{space} \end{cases}, \qquad \text{Eq. 5}$$

While the determined $Re$ is well within the laminar flow threshold, the measured RTD occurs earlier than theoretical laminar flow condition and exhibits a broaden distribution. The discrepancy between them indicates that the dispersions or potential secondary flows could dominate the flow regime. Instead, an improved Taylor dispersion model (shown as the gray dot-dash line in Figure 8) is able to reproduce the measured RTD, which was previously implemented in the characterization of photooxidation flow reactors (Lambe et al., 2011). Two flow patterns with distinct effective diffusivities (0.02 and 0.51 derived from best fit) were



considered in this dispersion model. An implication from the characteristics of the model is
that two flow components consist of the flow regime: a direct flow path through the flow tube
with less diffusion and a secondary flow path representing the recirculation in the dead zone
that induced by temperature gradient and significant diffusions (Huang et al., 2017).

### 494     4.3 $N_2O_5$ wall loss

The stainless-steel flow tube in this study is electro-polished and coated by FEP inside to
reduce the loss of $N_2O_5$ and particles on the wall in the meantime. An electro-polished surface
could enhance the homogeneity of FEP-coating and reduce the adsorption of $H_2O$ molecule to
the wall, which influences the loss of $N_2O_5$. It has been found that the $k_{wall}$ of $N_2O_5$ increases
with the RH (Bertram et al., 2009a;Wang et al., 2018c). Therefore, a less change in $k_{wall}$ of
$N_2O_5$ from RH helps to minimize the uncertainty induced by fluctuations of RH within a duty
cycle. Laboratory tests were conducted to quantify the $k_{wall}$ of $N_2O_5$ under different levels of
RH with HEPA inline. As shown in Figure 9, the $k_{wall}$ of $N_2O_5$ gradually increase from 0.002
$s^{-1}$ in a dry condition to 0.006 $s^{-1}$ when RH is 70%. The level of $k_{wall}$ is less than the result of
Wang et al. (2018c) but higher than Bertram et al. (2009a) as indicated in Table 2. In addition,
the flow tube was rinsed with deionized water every week during the field campaigns to
remove the build-up of particles, which might increase the hygroscopicity of the internal
surface and thus the $k_{wall}$ of $N_2O_5$ in a wet condition. Uncertainty in $\gamma(N_2O_5)$ derivation resulted
from the variation of $k_{wall}$ related to RH is discussed in section 5.

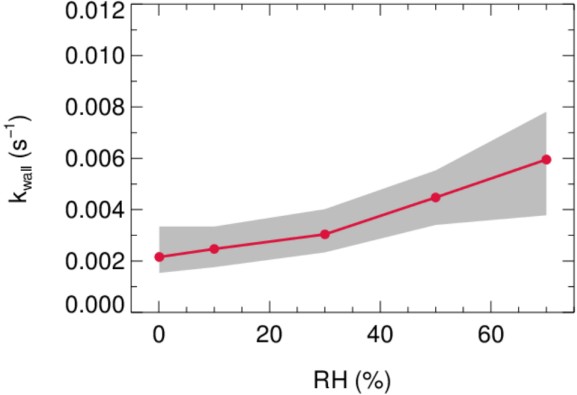


**Figure 9**. The dependence of pseudo-first-order wall loss coefficient ($k_{wall}$) of $N_2O_5$ in the





FEP-coated aerosol flow tube.
**Table 2**. Summary of the $k_{wall}$ of $N_2O_5$ for the existing aerosol flow tube deployed in field
campaigns.

| RH range | $k_{wall}$ range ($\times 10^{-3}$ s$^{-1}$) | References |
|----------|----------------------------------------------|------------|
| 5~50% | 0.5~3 | Bertram et al., 2009 |
| 20~70% | 4~9 | Wang et al., 2018 |
| 0~70% | 2~6 | This work |

## 4.4 Demonstration of γ(N₂O₅) measurements on model particles

γ(N₂O₅) measurements by current aerosol flow tube system equipped with box model method
were performed on lab-generated (NH₄)₂SO₄ aerosols over a range of RH. The system was
operated at room temperature of 295K with $N_2O_5$ concentration of 4.0 ppbv at the entrance of
flow tube. We conditioned the RH of generated aerosols by introducing dry $N_2$ gas dilution,
which could decrease the RH level down to 10~55%, starting from over 95% where (NH₄)₂SO₄
aerosols are expected to be in aqueous state. The resulting Sa concentrations of aerosols were
around 600 μm²·cm⁻³. As shown in Figure 10, the observed γ(N₂O₅) values were below 0.01
when RH was within 40% and significantly rose up to 0.02 with higher RH. The dependence
of γ(N₂O₅) on RH and the exact values are well consistent with previous laboratory results on
(NH₄)₂SO₄ aerosols (Badger et al., 2006;Hallquist et al., 2003;Hu and Abbatt, 1997;Kane et
al., 2001;Mozurkewich and Calvert, 1988), which shows that the setup of our instrument has
good practicability. A large standard deviation of γ(N₂O₅) found at RH of 39% is possibly due
to the unstable phase transition of (NH₄)₂SO₄ particles, as its efflorescence RH is reportedly
from 35 to 48% (Martin, 2000).



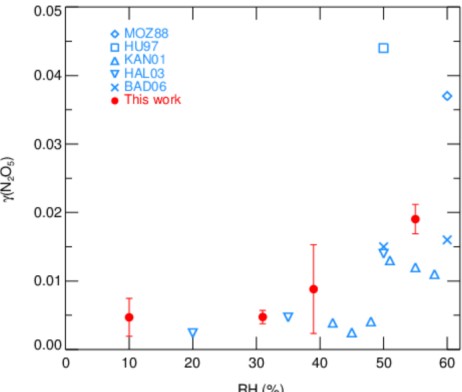


**Figure 10**. The dependence of $\gamma(N_2O_5)$ on RH for laboratory-generated $(NH_4)_2SO_4$ aerosols.
The red points with standard deviations represent the values measured by current aerosol flow
tube system in this work. Previously reported values are indicated in blue marks.

## 5 Uncertainty analysis and detection limit

The uncertainty of $\gamma(N_2O_5)$ is in relevance to the measurement uncertainties of each instrument
and rapid fluctuations of various parameters. As outlined before, the 5-min averages of $N_2O_5$
concentration measured at the inlet and exit of the flow tube were used for calculating $\gamma(N_2O_5)$
via the box model method. The potential variations within these selected time periods would
therefore lead to relative errors. For example, the variations of $N_2O_5$ concentration is resulted
majorly from the rapid changes of ambient NO and less from variations of VOCs, $NO_2$, $O_3$ as
well as $N_2O_5$ gas source itself (1% in 24 hours). A cutoff of 10% for $N_2O_5$ variation was
implemented to filter out the air mass that was too unstable for valid analysis, according to our
prescribed criteria of data screening. It consequently leads to 10% uncertainty in the average
of $N_2O_5$ and can translate into a deviation of 2% in $\gamma(N_2O_5)$ with the $\gamma(N_2O_5)$ at 0.02, Sa at
800 $\mu m^2 \cdot cm^{-3}$ and other parameters (shown in Table 3) representing the typical inlet values
measured during the field campaign (described in section 6). Similarly, cases that over 2%
variation in RH exists between the HEPA inline and bypass mode are excluded from analysis,
owing to its significant influence on $k_{wall}$ of $N_2O_5$ in the flow tube. By assuming a consistent
$k_{wall}$ in successive sampling modes, the potential variations in RH could lead to uncertainty in





$\gamma(N_2O_5)$ from $\pm 8 \times 10^{-4}$ at RH of 20% to $\pm 2 \times 10^{-3}$ at RH of 70%, respectively, with the Sa at
800 $\mu m^2\ cm^{-3}$. In addition, the $k_{NO3\text{-}VOCs}$ is treated as constant in a duty cycle due to the limit
of time resolution of VOCs measurements. A variation of $\pm 0.01\ s^{-1}$ in $k_{NO3\text{-}VOCs}$ only induces
less than $\pm 1\%$ uncertainty in $\gamma(N_2O_5)$ for more than 95% cases obtained during the field
campaign. All the impacts from inherent instruments uncertainties and variations of different
parameters are thereby considered in Monte Carlo simulations to assess the overall uncertainty
of $\gamma(N_2O_5)$. The basic simulation is initialized with the typical conditions measured at the inlet
of the flow tube during the field campaign and repeatedly performs the procedures of
determining $\gamma(N_2O_5)$ via the box model method 1000 times. In each run, all parameters were
allowed to vary independently within a prescribed range, presented in Table 3.
**Table 3.** Parameters involved in the Monte Carlo simulations.

| Parameters | Value [a] | Variation range [b] |
|---|---|---|
| NO | 1 ppbv | $\pm 10\%$ |
| $NO_2$ | 70 ppbv | $\pm 10\%$ |
| $O_3$ | 10 ppbv | $\pm 5\%$ |
| Inlet $N_2O_5$ | 4 ppbv | $\pm 19\%$ |
| Exit $N_2O_5$ [c] | 2.2 ppbv | $\pm 19\%$ |
| Temperature | 273 K | $\pm 0.1$ K |
| RH [d] | 30 % | $\pm 1\%$ |
| $k_{NO3\text{-}VOCs}$ | 0.01 $s^{-1}$ | $\pm 0.01\ s^{-1}$ |

[a] Values used for initializing Monte Carlo simulations in a basic scenario; [b] Ranges within
which each parameter can vary independently; [c] Determined from the case that $\gamma(N_2O_5)$ is at
0.02, Sa is at 800 $\mu m^2 \cdot cm^{-3}$ and other parameters are shown in this table; [d] The RH and its
variation can be transformed into values in $k_{wall}$ of $N_2O_5$ via the fitting function derived from
Figure 9.
The resulting $\gamma(N_2O_5)$ values from Monte Carlo simulations under the basic scenario are
shown as frequency distributions in Figure 11(a). This distribution can be fitted by a Gaussian
function and the standard deviation ($1\sigma$) of Gaussian distribution is regarded as the overall



uncertainty of $\gamma(N_2O_5)$, which is $\pm 9 \times 10^{-4}$ (4.5% relative to true $\gamma(N_2O_5)$). The uncertainty of
Sa measurements and unmeasured particles larger than 730 nm (usually less than 5% of total
Sa) would together introduce an extra 16% uncertainty to $\gamma(N_2O_5)$.

We further found that the uncertainty of $\gamma(N_2O_5)$ could be sensitive to the measurement

conditions. With higher $O_3$, potential variations of NO and $k_{NO3-VOCs}$ will induce larger
uncertainty of $\gamma(N_2O_5)$ (Figure 11(b)), as it enhances the abundance of $NO_3$ and $N_2O_5$. In
comparison, the low $O_3$ in the basic scenario suppressed the side formation of $NO_3$ in the flow
tube, limiting the aggravation of $\gamma(N_2O_5)$ uncertainty from the increase of NO and $NO_2$. The
$\gamma(N_2O_5)$ uncertainty is also positive correlated with RH and T. As is discussed before, the $k_{wall}$
of $N_2O_5$ increases with RH level, which can amplify the potential bias of $k_{wall}$ at a higher RH
level. The equilibrium between $NO_3$ and $N_2O_5$ shifts towards the decomposition of $N_2O_5$ at
higher T, leading to larger uncertainty of $\gamma(N_2O_5)$ caused by potential variations of NO and
$k_{NO3-VOCs}$. The overall uncertainty of $\gamma(N_2O_5)$ therefore rises to 8.2% at the RH of 70% and to
14.4% at the temperature of 293K (Figure 11(c)), with NO, $NO_2$, $O_3$, $\gamma(N_2O_5)$ and Sa keeping
the same as the basic scenario. In addition, Monte Carlo simulations were also performed for
different $\gamma(N_2O_5)$ values ranging from 0.01 to 0.08. The uncertainty of $\gamma(N_2O_5)$ clearly
decreased with the $\gamma(N_2O_5)$ (Figure 11(d)). A lower $\gamma(N_2O_5)$ weaken the impacts $N_2O_5$ uptakes
has on the budgets of $NO_3$ and $N_2O_5$, which causes the $\gamma(N_2O_5)$ derivation to be more
susceptible to uncertainties of other parameters and then increases the uncertainty of $\gamma(N_2O_5)$.



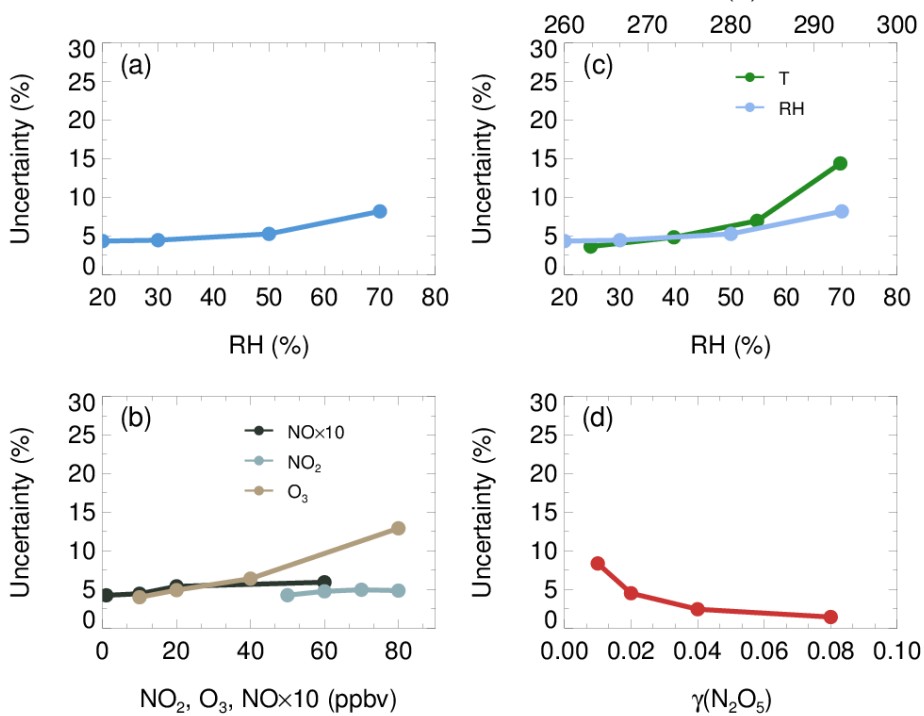

**Figure 11.** The uncertainty of $\gamma(N_2O_5)$ determined from the Monte Carlo simulations. (a) Histogram distribution of $\gamma(N_2O_5)$ generated from a Monte Carlo simulation (1000 single runs) in the basic scenario (shown as Table 2), where the overall uncertainty of $\gamma(N_2O_5)$ was determined to be $\pm 9 \times 10^{-4}$; (b) dependence of the uncertainty of $\gamma(N_2O_5)$ on NO, $NO_2$ as well as $O_3$; (c) dependence of the uncertainty of $\gamma(N_2O_5)$ on RH and T; (d) dependence of the $\gamma(N_2O_5)$ uncertainty on $\gamma(N_2O_5)$ level.

In addition, the mean residence time used in the box model method could bias the retrieved $\gamma(N_2O_5)$ due to the non-normal distribution of residence time with a discernable tail. The reactants entrained by those slower streamlines close to the wall will take much longer time to reach the exit of the flow tube than that by the centerline. In order to evaluate the uncertainty caused by the distribution of residence time, we first performed simulations of $N_2O_5$ decay in the flow tube under the basic scenarios and calculate the exit $N_2O_5$ concentration according to the probability distribution function derived from RTD profile. Then the $\gamma(N_2O_5)$ can be retrieved from the box model method running for the duration of mean residence time,

constrained by this calculated exit $N_2O_5$ concentration. The result shows an underestimation
of $\gamma(N_2O_5)$ derived from the mean residence time reaching 32% in the basic scenario. The
extent of underestimation is most sensitive to the level of $\gamma(N_2O_5)$ and RH. In short, when
taking all the factors and their corresponding varying ranges discussed above into
consideration, the overall uncertainty of $\gamma(N_2O_5)$ determined from Monte Carlo simulations is
in the range of 16-74%.

In order to determine the detection limit of the current aerosol tube system, the continuous

blank measurements in zero air were performed with settled operation procedures. Within per
duty cycle (40 minutes), one $k_{wall}$ of $N_2O_5$ and one $\gamma(N_2O_5)$ can be derived in pair. In total, we
obtained 56 sets of result. The detection limit of $k_{N2O5}$ on aerosols is $2.1 \times 10^{-5}$ $s^{-1}$, derived from
$1\sigma$ of the Gaussian function fitted to this distribution. It is equivalent to 0.0016 for the detection
limit of $\gamma(N_2O_5)$ with a low Sa condition of 200 $\mu m^2$ $cm^{-3}$ (Figure 12(a)), and 0.00064 for the
detection limit of $\gamma(N_2O_5)$ with a moderate Sa condition of 500 $\mu m^2$ $cm^{-3}$ (Figure 12(b)). This
result indicates that the flow tube system has capability of quantifying $\gamma(N_2O_5)$ for most cases
even under a low aerosol-loading environment.

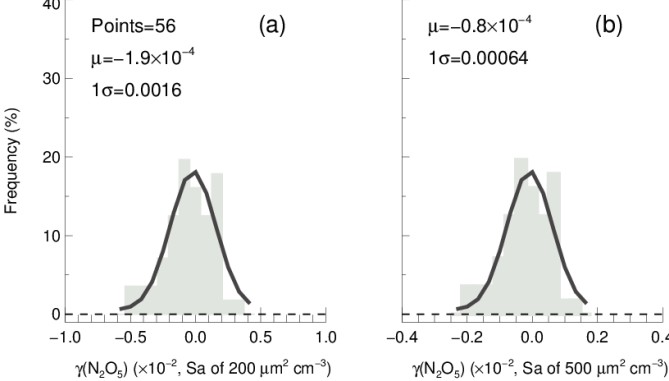


**Figure 12**. The $\gamma(N_2O_5)$ derived from blank measurements in histogram distribution plot. The
$\gamma(N_2O_5)$ was calculated from $k_{N2O5}$ by Eq 2 with Sa of (a) 200 $\mu m^2$ $cm^{-3}$ and (b) 500 $\mu m^2$ $cm^{-3}$,
respectively, under the temperature of 293K. The Gaussian function is fitted to the distribution
and plotted in black line. The $1\sigma$ from Gaussian fit is regarded as the detection limit.



## 6 Performance in the field campaign


The aerosol flow tube system was successfully deployed to measure $\gamma(N_2O_5)$ on ambient
aerosols in Beijing lasting for 20 days during the December of 2020. The sampling site was at
the campus of Peking University, which is located in the city center of Beijing surrounded by
major roads with heavy traffic. Therefore, this site represents an area with large amount of
fresh emission of NOx and other anthropogenic sources. The system was mounted in the top
floor of a building, about 15 m height above the ground. The sampling manifold was placed
in open air and the ambient aerosols could directly enter the inlet of the manifold without
additional sampling tubes. During the period of measurement, the averages of ambient
temperature, RH, NO, $NO_2$, $O_3$ and Sa were $273 \pm 3$ K, $25 \pm 12$ %, $23 \pm 36$ ppbv, $23 \pm 12$ ppbv,
$16 \pm 15$ ppbv and $409 \pm 249$ $\mu m^2$ $cm^{-3}$, respectively. The NO and Sa levels could vary by 2
orders of magnitude due to the periodical switch between clean air mass from the north and
pollutants accumulated by local emission.
A total of 99 valid $\gamma(N_2O_5)$ values were determined from the measurements based on the
criteria of data screening described in section 3.1. We found that $\gamma(N_2O_5)$ was $0.042 \pm 0.026$ on
average with a median of 0.035, ranging from 0.0045 to 0.12 (Figure 13). These results are
comparable to that previously determined in the North of China using various different
methods (Wang et al., 2017b;Wang et al., 2018b;Wang et al., 2017d;Wang et al., 2017e;Xia et
al., 2019;Yu et al., 2020a). The $k_{wall}$ of $N_2O_5$ corresponding to valid $\gamma(N_2O_5)$ measurements
was rather stable at an average of $0.0021 \pm 0.0007$ $s^{-1}$, which was consistent with the values
determined at similar RH levels in the laboratory tests. It somehow reflected the robustness of
the status of the flow tube system and the derived results.
In the current system, the $N_2O_5$ concentrations measured at both entrance and exit of the
flow tube are sensitive to the NO fluctuations within the timescale of one sampling mode,
which can induce large uncertainty on calculating $\gamma(N_2O_5)$. With our stringent criteria of data
screening, the cases of drastic NO fluctuations were excluded from the analysis. Hence, the
majority of valid $\gamma(N_2O_5)$ for this campaign were obtained during the periods of the NO below
2 ppbv, when the clean air mass was dominant at this urban site. Meanwhile, the Sa



concentration within clean episodes were also lower than other periods, with an average of
159 $\mu m^2 \, cm^{-3}$. The derived $k_{N2O5}$ ranged from $2.1 \times 10^{-5}$ to $1.6 \times 10^{-3}$ $s^{-1}$ well above the
detection limit, which demonstrated the robustness of results even subject to low ambient Sa
conditions. In order to improve the applicability of $\gamma(N_2O_5)$ measurements, future
development is suggested to prioritize the reduction or removal of NO level (at least the
fluctuation of NO) in the sampling system before the entrance of flow tube without the cost of
particles transmission efficiency.

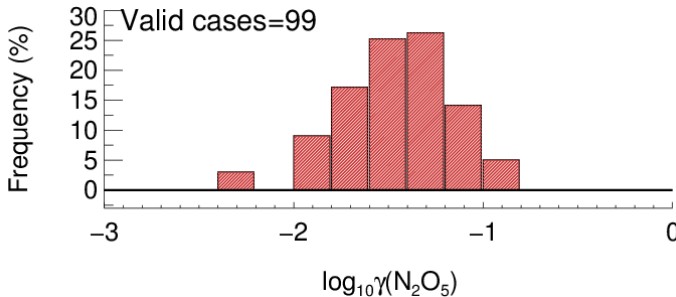


**Figure 13**. The histogram distribution of measured $\gamma(N_2O_5)$ for valid cases.
**7 Summary and conclusion**
We report a new development of an aerosol flow tube system coupled with detailed box model
to derive $\gamma(N_2O_5)$ directly on ambient aerosols. The unique feature of this system is that the
simultaneous $N_2O_5$ measurement at the both ends of flow tube was applied to improve the
accuracy in quantifying $\gamma(N_2O_5)$, by taking it as a constraint for the box model to reproduce
the decay of introduced $N_2O_5$ gas source in the flow tube. With the consideration of detailed
chemistry related to $N_2O_5$, the proposed approach was testified to refrain from the interference
of side reactions under different conditions, induced by the additional $N_2O_5$ generation, NO
titration in the flow tube and variations of air masses between successive sampling modes.

A series of laboratory tests were performed to characterize factors affecting $\gamma(N_2O_5)$

derivation and demonstrate its applicability on $(NH_4)_2SO_4$ aerosols. The uncertainties
associated with instruments used in the system and potential fluctuations of various parameters
were thoroughly discussed in the uncertainty analysis, and we estimated the overall uncertainty





of $\gamma(N_2O_5)$ to be 16-74% which is subject to NO, $NO_2$, $O_3$, meteorological parameters,
residence time and $\gamma(N_2O_5)$ value itself. The detection limit of $\gamma(N_2O_5)$ was quantified to be
0.0016 at the aerosol surface concentration (Sa) of 200 $\mu m^2$ $cm^{-3}$. We deployed this system for
field observations of $\gamma(N_2O_5)$ at an urban site in Beijing, where strong anthropogenic emission
and periodically switch of air mass were encountered. The obtained $\gamma(N_2O_5)$ was in
comparable level to previously reported values in the north of China and demonstrated the
robustness of this system within low NO episodes. Further investigations on $N_2O_5$
heterogeneous chemistry for both laboratory-generated and ambient particles are also
available by the introduced approach.



**Code/Data availability.** The datasets used in this study are available from the corresponding
author upon request (wanghch27@mail.sysu.edu.cn; k.lu@pku.edu.cn).

**Author contributions.** K.D.L. and H.C.W. designed the study. X.R.C and H.C.W. analyzed
the data and wrote the paper with input from K.D.L.

**Competing interests**. The authors declare that they have no conflicts of interest.

**Acknowledgments**. This project is supported by the National Natural Science Foundation of
China (21976006, 42175111); the Beijing Municipal Natural Science Foundation for
Distinguished Young Scholars (JQ19031); National State Environmental Protection Key
Laboratory of Formation and Prevention of Urban Air Pollution Complex (CX2020080578);
the special fund of the State Key Joint Laboratory of Environment Simulation and Pollution
Control (21K02ESPCP); the National Research Program for Key Issue in Air Pollution
Control (DQGG0103-01, 2019YFC0214800). Thanks for the data contributed by field
campaign team.

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
