# Peer review of "Direct measurement of N2O5 heterogeneous uptake coefficients on ambient"

_Atmospheric Measurement Techniques, 2022_

## Author Comment (AC1)

**Reply on RC1**

We appreciate the reviewer for the careful reading and their constructive comments on our manuscript. **As detailed below, the reviewer's comments are** normal font, **our response to the comments are shown as** *italicized font*. **New or modified text is in blue.**

All the line numbers refer to original version of Manuscript ID: **amt-2022-167**.

This paper provides a modification to ambient flow tube systems developed by Bertram et al 2009 and Wang et al 2018. The authors document the increased robustness due to measurements prior and after the flow tube as well as a box model to predict side reactions. While this is an interesting finding, I do not think this work is novel enough for publication in AMT. This work belongs as a technical note. I also noted that the authors need to credit Bertram et al 2009 more for use of their design. My comments are below.

*We thank this comment. The reviewer emphasized the similarity of our work to the two previous works and questioned the novelty of this study. While our work builds on the previous two studies and attempts to address some issues that were not well addressed in the above work.*

*First, accurate quantification of wall loss is essential for the quantification of $N_2O_5$ uptake coefficients as suggested in Bertram's paper, a better way to reduce the wall loss interference is to measure it frequently (Bertram et al., 2009, see section 4.3). This suggestion was subsequently adopted by Wang et al. but the concentration of the $N_2O_5$ source they used in determining the wall loss and the total $N_2O_5$ loss was an assumed stable value rather than an observed one. Here, we dynamically determine the $N_2O_5$ source concentration frequently, which is helpful to provide accurate data on quantifying the wall loss as well as the total $N_2O_5$ loss in the flow tube in each measurement cycle. Figure R1 shows the example of difference of measured $N_2O_5$ wall loss at lab condition and ambient condition.*

[Figure]

*Figure R1. The derived dependence of $N_2O_5$ wall loss on RH at laboratory condition (red dots) and field measurement (blue square)*

*Second, the use of an iterative box model approach to correct the potential bias, due to side reactions in the flow tube, is the highlight of the work of Wang et al. The iterative box model calculates backward to estimate the concentration of $NO_2$ and $O3$ before entering the flow tube, **with estimated NO profile and measured $N_2O_5$ at the exit of flow tube.** Here, the observed concentrations of $NO_2$, $O3$, and NO at "time zero" are obtained through programmed cyclic measurements, which can reduce the uncertainties by adding the model constrain (more details can be found in the reply for the following Question NO. 4). Figure R2(a) presents the box whisker of $N_2O_5$ and NO concentration before the entrance during a field campaign. The variation of initial $N_2O_5$ is much larger than that in lab condition with a very small standard deviation for $N_2O_5$ concentration (<1%, mentioned in line 180), highlights the influence of ambient air mass (e.g. varying NO and temperature) to injected $N_2O_5$. Figure 2R(b) shows the case of large underestimation by using a fixed initial $N_2O_5$ concentration rather than measured values as box model input. In addition, we simulate NO3-$N_2O_5$ relationship via specific reactions rather than approximating it in equilibrium and introducing the equilibrium coefficient (Keq) into calculation. Determining NO3 or $N_2O_5$ concentration by Keq could induce large bias (up to 90%) under the high aerosol loading and low temperature (Chen et al., 2021).*

[Figure]

*Figure R2. (a) the box whisker of $N_2O_5$ source and NO measured before the entrance; (b) the inter-comparison of derived $N_2O_5$ uptake coefficient by using a fixed $N_2O_5$ and a dynamic measured $N_2O_5$ at the time zero in the iterative box model.*

*Third, to achieve the programmed cyclic measurement of key parameters such as $N_2O_5$, NOx and O3, we adopted a new design scheme of Y-tee and cyclic measurement setup, which were never used in the previous two studies.*

*In summary, we believe that the above innovation points of our study advance the field measurement of $N_2O_5$ uptake coefficient and be enough for publication in AMT, while we may not address our innovation points well in the previous version. Thus, we have added an explanation of the novelty, and credited Bertram et al 2009 and Wang*

*et al 2018 more for use of their design in the revised manuscript. The detailed revision can be found in the following point to point responses.*

MAJOR COMMENTS AND CONCERNS

1. Sections 2.1-2.3, the bulk of the flow tube design, are all pretty much the same as (T. H. . Bertram et al., 2009) yet the authors do not cite this paper in these sections for the design. A reader who has not read Bertram et al, 2009 might very well think that these aspects of the design are the authors'.

*We thank the reviewer for pointing out this issue. We have added the citation of this paper and added the statements to clarify the similarity and difference between Bertram et al. (2009) and this work in the beginning of section 2 and section 2.3. The description is modified as follows:*

"The design of sampling module and aerosol flow tube in this work follows previous work for measuring $\gamma(N_2O_5)$ on ambient aerosols (e.g. Bertram et al., 2009). The major distinctions of this system from previous work are continuous monitor of NOx and $O_3$ concentration before the inlet of flow tube (after sampling air mixing with $N_2O_5$ source) and the sequential measurements of $N_2O_5$ concentration both at the inlet and the exit of flow tube within a duty cycle."

"The mechanic design of this flow tube follows that used in Bertram et al. (2009), with different length and diffuser angles particularly designed for our typical flow rate."

2. The residence time in the flow tube is quoted at 156s, which seems way too short for low values of surface area.

*Thank you for pointing out this issue and we totally understand the reviewer's concern. In fact, we mainly focus on the typical episode days with medium to high aerosol loadings (the surface area concentration larger than 500 $um^2/cm^3$) in polluted regions, such as Northern China. As indicated in section 5, under the Sa of 500 $um^2/cm^3$, the detection limit of this system is $6.4 \times 10^{-4}$, which is well below the previous most of previous ambient $\gamma(N_2O_5)$ measurement results ranging from $1 \times 10^{-3}$ to >0.1 in polluted regions of China(Wang et al., 2020;Wang et al., 2017c;Wang et al., 2017d;Xia et al., 2019). The residence time determined in this work is also slightly higher than 149 s that reported in a previous work focusing on investigating $\gamma(N_2O_5)$ in polluted regions(Wang et al., 2018b). In addition, the residence time for our system can be extended to over 300 s by reducing the air flow rate inside the flow tube. The flow rate of current set up is controlled by both the detection instruments and an extra pump with 1 SLPM flow rate attached to the bottom of the flow tube, which was actually designed for adjusting the residence time of the flow tube in order*

*to match different levels of aerosol loadings. We believe this additional set up can satisfy the requirement of ambient γ(N₂O₅) measurement and research purpose of N₂O₅ uptake under low surface area concentration. More clarifications have been added in section 4.2 as follows.*

"As shown in Section 5, the detection limit of this system is $6.4 \times 10^{-4}$ with Sa of 500 $um^2$ $cm^{-3}$, which is well below the previous most of previous ambient $\gamma(N_2O_5)$ measurement results ranging from $1 \times 10^{-3}$ to >0.1 in polluted regions of China (Wang et al., 2020;Wang et al., 2017c;Wang et al., 2017d;Xia et al., 2019). The residence time determined in this work is also slightly higher than 149 s that reported in a previous work focusing on investigating $\gamma(N_2O_5)$ in polluted regions(Wang et al., 2018b). In addition, the residence time for this flow tube can be extended to over 300 s to satisfy the $\gamma(N_2O_5)$ measurement requirements under low Sa by reducing the flow rate of air passing through, which is controlled by an extra pump."

3. The authors measure very high values of gamma, much higher than observed for most ambient studies: (T. H. Bertram et al., 2009; Riedel et al., 2012b) and most flow tube work with the exception of dust particles (Mitroo et al., 2019; Tang et al., 2016), I wonder if the authors have underestimated the particle surface area by not using an APS or if the filter upstream of the CEAS is causing an artificially higher gamma than expected.

*Thank you for pointing out this issue. The γ(N₂O₅) measured in this work ranged from 0.0045 to 0.12, which was within the range of $10^{-5}$ to >0.1 determined from ambient N₂O₅ measurements around the world (Bertram et al., 2009;Brown et al., 2009;McDuffie et al., 2018;Morgan et al., 2015;Tham et al., 2018) and comparable to previous results reported in polluted regions in China (Wang et al., 2020;Wang et al., 2018a;Wang et al., 2017c;Wang et al., 2017d;Xia et al., 2019;Yu et al., 2020). Although it is true that the average of γ(N₂O₅) (0.042) measured in Beijing 2020 was somewhat higher than other reported average values, this phenomenon is majorly associated with the air mass encountered during measurement period. Owing to the data filtering criterion applied in this study, the valid γ(N₂O₅) was mostly measured within the air mass came from the North of Beijing with low NO and VOCs concentration (see the following Figure R3). The simultaneous aerosol component measurements by ACSM and elements distribution analysis for single particle by TEM (Transmission electron microscope) indicated that the organics accounted for less than 50% of aerosol dry mass and organic coating was not prevalent on aerosols, respectively, during the periods with valid γ(N₂O₅) data (will be presented in a following paper and see also the following Figure R4). These evidences may explain the high γ(N₂O₅) values measured in this study.*

[Figure]

*Figure R3. Time series of in-situ measured γ(N₂O₅) and NO concentration during Dec 09~21 in 2020.*

[Figure]

*Figure R4. The relation plot of γ(N₂O₅) and organic dry mass fraction, color coded with aerosol water content.*

*For the possibility of γ(N₂O₅) underestimation mentioned by the reviewer, we have included the possible underestimation of Sa and the filter loss of N₂O₅ in the uncertainty analysis. According to on-site characterization of particle size distribution (shown in section 4.1), the Sa concentration distribution peaked at 200-300 nm and the particles larger than 730 nm was estimated to account for less than 5% of Sa concentration. The membrane filter was changed every 2 hours during the measurement period and the transmission efficiency of N₂O₅ on the used membrane was determined to be 92 ±3% (also see section 2.4). Such an 8% loss of N₂O₅ was corrected in N₂O₅ measurement. Overall, 5% uncertainty of Sa measurement and 19% uncertainty of N₂O₅ measurement (with 3% resulted from N₂O₅ loss on membrane filter) have been considered in the total uncertainty of γ(N₂O₅) as indicated in section 5.*

4. It is not clear that the box model presented in this work is a significant advance over the box model presented in Bertram et al 2009 and Wang 2018. The authors need to provide evidence that their work is a significant advance over previous work.

*We gratefully appreciate for your valuable suggestion. The box model presented in this work can quantify the wall loss rate constant of $N_2O_5$ and $\gamma(N_2O_5)$, respectively, in one duty cycle. It particularly considers the variation of gas-phase reactions rates and the equilibrium between $NO_3$ and $N_2O_5$, with the constraint of $N_2O_5$ concentration measured sequentially at both end of the flow tube. In comparison with Bertram et al 2009 and Wang et al 2018, the method we retrieved $\gamma(N_2O_5)$ by using the box model in this work has advances in avoiding the influence resulted from varying air mass (such as varying NOx, $O_3$ and RH level), from the equilibrium approximation used in iterative box model and from the variation of initial $N_2O_5$ concentration.*

*In the study of Bertram et al 2009, they used the box model to check for the concentration of $N_2O_5$ source and the uncertainty from gas phase reactions in some cases. Routinely, $\gamma(N_2O_5)$ was retrieved directly by the $N_2O_5$ concentration measured at the exit of flow tube with and without particles inline. In this work, we used the box model to retrieve $\gamma(N_2O_5)$ based on measurements of NOx, $O_3$ and $N_2O_5$ from time to time. We found that our method can avoid the underestimation of $\gamma(N_2O_5)$ derived by exit-concentration method under different level of NO, $NO_2$ and $O_3$ due to the lack of consideration of gas phase reaction. Simulations and laboratory tests on $(NH_4)_2SO_4$ aerosols presented in figure 6 and section 3.2 corroborated that our method can buffer the variation of air mass and the resulted bias from the changes of gas phase reaction rates.*

*In the study of Wang et al 2018, they used an iterative box model to retrieve $\gamma(N_2O_5)$ based on measurements of NOx, $O_3$ and $N_2O_5$ only at the exit of the flow tube. There are two basic assumptions within the simulation of the iterative box model: The first one is that $NO_3$ and $N_2O_5$ are always in equilibrium and equilibrium coefficient Keq can be represented by simple parameterization to derive the NO profile; The second one is that initial $N_2O_5$ is stable during field measurements. However, in our previous study, the simple parameterization of Keq was found not applicable under high aerosol loading or low temperature, which could lead to over 90% overestimation on Keq value in polluted episode days (Chen et al., 2022). In addition, the initial $N_2O_5$ concentration after mixing with sampling air is not expected to be as stable as that observed in laboratory tests, due to the variations in temperature, NO concentration and other related parameters. Therefore, the direct constraint of initial $N_2O_5$, NOx and $O_3$ concentration measured at the inlet of the flow tube in this work enable a straightforward simulation of $NO_3$-$N_2O_5$ chemistry occurring in the flow tube, which reduces the uncertainty of $\gamma(N_2O_5)$ derivation. We have rephrased the statement about the advances of model used in this work and presented it in section 3.*

SPECIFIC COMMENTS

Abstract

1. "newly developed" on line 19. From the paper, this just seems like a slight modification instead.

*Thanks for your comment. We have rephrased it as follows.*

"An improved aerosol flow tube system coupled with detailed box model was developed to measure $N_2O_5$ heterogeneous uptake coefficients ($\gamma(N_2O_5)$) on ambient aerosols directly."

Introduction

2. Line 47, also cite (Gaston and Thornton, 2016; Mitroo et al., 2019; Riedel et al., 2012a, 2013)

*Thanks for your suggestion. We have cited the references as mentioned by the reviewer.*

3. Line 59, also cite (Cosman et al., 2008; Escorcia et al., 2010; Folkers et al., 2003; Gaston et al., 2014)

*Thanks for your suggestion. We have cited the references as mentioned by the reviewer.*

4. Lines 62-65, also mention particle size as well (Gaston and Thornton, 2016). Also missing papers on organic aerosol (Escorcia et al., 2010; Gaston et al., 2014; Griffiths et al., 2009; Thornton et al., 2003)

*Thanks for your valuable suggestion. We have added the particle size as well and cited the papers on organic aerosol mentioned by the reviewer.*

5. Lines 71-74 reflect the findings in (Thornton et al., 2003), which should be cited here.

*Thanks for your careful check and suggestion. We have cited the reference as mentioned by the reviewer.*

6. Lines 74-77 reflect the findings in (Mitroo et al., 2019; Royer et al., 2021), which should be cited here.

*Thanks for your careful check and suggestion. We have cited the references as mentioned by the reviewer.*

7. Line 105, Mitroo et al 2019 did not use an ambient flow tube.

*We are sorry for the incorrect citing and have removed Mitroo et al 2019 here.*

Methods

1. Sections 2.1-2.3 are really the design of (T. H. . Bertram et al., 2009; T. H. Bertram et al., 2009), as such, the authors must use appropriate citations here.

*Thanks for your suggestion. We have added the citation as mentioned by the reviewer here and rephrased the description of our design to distinguish from that in Bertram et al 2009. The description is modified as follows:*

"The design of sampling module and aerosol flow tube in this work follows previous work for measuring $\gamma(N_2O_5)$ on ambient aerosols (e.g. Bertram et al., 2009). The major distinctions of this system from previous work are continuous monitor of NOx and $O_3$ concentration before the inlet of flow tube (after sampling air mixing with $N_2O_5$ source) and the sequential measurements of $N_2O_5$ concentration both at the inlet and the exit of flow tube within a duty cycle."

"The mechanic design of this flow tube follows that used in Bertram et al. (2009), with different length and diffuser angles particularly designed for our typical flow rate."

2. Lines 215-217, wouldn't the use of a filter upstream of the CEAS cause issues where wet, ambient particles would react with $N_2O_5$ going into the CEAS and cause a higher gamma than one would expect? That might explain the very high values of gamma observed in ambient.

*We totally understand the reviewer's concern. The membrane filter was changed every 2 hours during the measurement period and the transmission efficiency of $N_2O_5$ on the used membrane was determined to be 92±3% (also see section 2.4). Such an 8% loss of $N_2O_5$ was corrected in $N_2O_5$ measurement and 3% uncertainty of of $N_2O_5$ measurement were also included in the uncertainty analysis of $\gamma(N_2O_5)$. The use of a filter upstream of the CEAS and the procedures of membrane changing have been successfully applied in many field campaigns to measure ambient $N_2O_5$ (Brown et al., 2016;Kennedy et al., 2011;Wang et al., 2017a;Wang et al., 2017b;Wang et al., 2018a). We further add the clarifications on the corrections for $N_2O_5$ possible loss on filter and cited references to support its applicability on $N_2O_5$ measurements in section 2.4. The modified texts are as follows.*

"Laboratory tests have been conducted to quantified the transmission efficiency of $N_2O_5$ over the membrane (92±3%), sampling tube of CEAS (99.7%) and the inside of CEAS (93.6%). The use of a filter upstream of the CEAS and the procedures of membrane changing have been successfully applied in many field campaigns to

measure ambient $N_2O_5$ (Brown et al., 2016;Kennedy et al., 2011;Wang et al., 2017a;Wang et al., 2017b;Wang et al., 2018a). The loss of $N_2O_5$ on membrane filter, sampling tube and the detection chamber inside the CEAS were corrected according to transmission efficiency and the detection limit of $N_2O_5$ was determined to be 2.7 pptv ($1\sigma$, 60s) with the measurement uncertainty of 19%."

*For the high gamma measured in this work, this phenomenon is majorly associated with the air mass encountered during measurement period. Please see also our response to comment 3 in major comment.*

3. Lines 287-288, this duty cycle is not that different from (T. H. . Bertram et al., 2009)

*Thanks for your suggestion. We have rephrased the description of duty cycle in this work and cited the work by Bertram et al 2009. The modified texts are as follows.*

"As a result, a typical duration of duty cycle is composed of 40 minutes with 20 minutes for each mode, which is similar to that in Bertram et al. (2009)."

4. Lines 305-308, what fraction of VOCs measured had known rate constants that can be used to parameterize the reaction of $NO_3$ with VOCs?

*A total of 59 kinds of VOCs were measured by GC-FID-MS in this work, half of which had known rate constants that can be used to parameterize the reaction of $NO_3$ with VOCs (mainly compose of alkenes and aromatics). Their rate constants were obtained from MCM331 or IUPAC and the values at 298K are listed as follows. We also added clarification about the measured VOCs with known rate constants in section 3.1 and the following table in the appendix.*

Table A1. VOCs used to calculate $NO_3$ reactivity ($k_{NO3}$) in the box model method

| Species | $k_{NO3}$(298 K) | Species | $k_{NO3}$(298 K) |
|---|---|---|---|
| METHANE | 1D-18 [b] | TRANS-2-PENTENE | 3.70D-13 [a] |
| ETHANE | 1D-17 [b] | 1-HEXENE | 1.20D-14 [a] |
| PROPANE | 7D-17 [b] | 1-3 BUTADIENE | 1.03D-13 [a] |
| N-BUTANE | 4.6D-17 [b] | ISOPRENE | 7.0D-13 [b] |
| I-BUTANE | 1.1D-16 [b] | STYRENE | 1.50D-12 [a] |
| ETHYLENE | 2.1D-16 [b] | ETHYNE | 1D-16 [b] |
| PROPYLENE | 9.5D-15 [b] | BENZENE | 3D-17 [b] |
| 1-BUTENE | 1.3D-14 [b] | TOLUENE | 7.8D-17 [b] |
| CIS-2-BUTENE | 3.50D-13 [a] | O-XYLENE | 4.10D-16 [a] |
| TRANS-2-BUTENE | 3.90D-13 [a] | M-XYLENE | 2.60D-16 [a] |

| I-BUTENE | 3.4D-13 [b] | P-XYLENE | 5.00D-16 [a] |
| 1-PENTENE | 1.20D-14 [a] | ETHYL BENZENE | 1.20D-16 [a] |
| CIS-2-PENTENE | 3.70D-13 [a] | N-PROPYL BENZENE | 1.40D-16 [a] |

Note: a. MCM; b. IUPAC

"The reaction of VOCs and $NO_3$ is treated as pseudo-first-order with a rate constant of $k_{NO3\text{-}VOCs}$, which is the sum of rate constants for reactions of $NO_3$ with each VOCs scaled by the concentration of VOCs measured by GC-FID. In this work, there are 30 kinds of measured VOCs having known reaction rate constants with $NO_3$ included in the model."

5. Section 3.2, it's not clear how this box model differs from the previous studies cited.

*Thanks for your comment. The model we used to retrieve $\gamma(N_2O_5)$ in this work was directly constrained by $N_2O_5$, NOx and $O_3$ concentration at the inlet of the flow tube, as well as $N_2O_5$ and Sa at the exit of the flow tube. In this method, the uncertainty of $\gamma(N_2O_5)$ derivation can be reduced owing to avoiding the influence resulted from varying air mass (such as varying NOx, $O_3$ and RH level), from the equilibrium approximation used in iterative box model and from the variation of initial $N_2O_5$ concentration. Please see also our response to comment 4 in major comments for more details as well.*

6. Lines 437-441 is similar to the findings of (T. H. Bertram et al., 2009)

*Cited the Bertram et al 2009 as follows.*

"The results we obtained from above particle transmission experiments are similar to the findings of Bertram et al. (2009)."

7. Lines 489-490, 156 s for a residence time is quite short, especially for low surface areas. What is the time required for complete mixing of $N_2O_5$ in the flow tube?

*Thanks for pointing out this issue. For the question of short residence time, please see also our response to comment 2 in major comments. For the question of the time required for complete mixing of $N_2O_5$ in the flow tube, we conducted experiments sampling $(NH_4)_2SO_4$ aerosols continuously. The measurement of $N_2O_5$ and Sa at the exit of the flow tube show that it took about 15 minutes for $N_2O_5$ and Sa to completely mix in the flow tube (as shown in the following figure). The residence time distribution (RTD) profiles (see in section 4.2) also demonstrated that a pulse injection of $NO_2$ requires 10~15 minutes to be fully released from the flow tube, which to some extent supports the 15-minute time required for complete mixing of $N_2O_5$. We have reorganized the statements on these results and presented it in section 2.5 as follows.*

"We measured Sa and $N_2O_5$ concentration continuously at the exit of flow tube when sampling $(NH_4)_2SO_4$ aerosols. As shown in the following figure, it took about 15 minutes for particles to rise to a stable level from none or to decrease from a certain level to none, when our system underwent mode switches. The correspondingly periodical variation of $N_2O_5$ concentration was consistent with particles. The residence time distribution (RTD) profiles (see in section 4.2) also demonstrated that a pulse injection of $NO_2$ requires 10~15 minutes to be fully released from the flow tube, which to some extent supports the 15-minute time required for complete mixing of $N_2O_5$."

[Figure]

Figure 3. Variations of Sa and $N_2O_5$ concentration (normalized to peak values) measured at the exit of flow tube when switching the sampling mode. The phases of species concentrations in the flow tube approaching stable after a mode switch are denoted as the transition phases.

8. Lines 635-639, gamma values seem really high. The authors should provide some explanation of how gamma varied as a function of air mass encountered.

*Thanks for pointing out this issue and your valuable suggestion. Please see also our response to comment 3 in major comments.*

**References:**

Bertram, T. H., Thornton, J. A., Riedel, T. P., Middlebrook, A. M., Bahreini, R., Bates, T. S., Quinn, P. K., and Coffman, D. J.: Direct observations of N2O5 reactivity on ambient aerosol particles, Geophys. Res. Lett., 36, 10.1029/2009gl040248, 2009.

Brown, S. S., Dube, W. P., Fuchs, H., Ryerson, T. B., Wollny, A. G., Brock, C. A., Bahreini, R., Middlebrook, A. M., Neuman, J. A., Atlas, E., Roberts, J. M., Osthoff, H. D., Trainer, M., Fehsenfeld, F. C., and Ravishankara, A. R.: Reactive uptake coefficients for N2O5 determined from aircraft measurements during the Second Texas Air Quality Study: Comparison to current model parameterizations, J. Geophys. Res.- Atmos., 114, D00F10(01-16), Artn D00f10 10.1029/2008jd011679, 2009.

Brown, S. S., Dubé, W. P., Tham, Y. J., Zha, Q., Xue, L., Poon, S., Wang, Z., Blake, D. R., Tsui, W., Parrish, D. D., and Wang, T.: Nighttime chemistry at a high altitude site above Hong Kong, J. Geophys. Res.: Atmos., 121, 2457-2475, 10.1002/2015jd024566, 2016.

Chen, X. R., Wang, H. C., and Lu, K. D.: Interpretation of NO3-N2O5 observation via steady state in high-aerosol air mass: the impact of equilibrium coefficient in ambient conditions, Atmos. Chem. Phys., 22, 3525-3533, 10.5194/acp-22-3525-2022, 2022.

Kennedy, O. J., Ouyang, B., Langridge, J. M., Daniels, M. J. S., Bauguitte, S., Freshwater, R., McLeod, M. W., Ironmonger, C., Sendall, J., Norris, O., Nightingale, R., Ball, S. M., and Jones, R. L.: An aircraft based three channel broadband cavity enhanced absorption spectrometer for simultaneous measurements of NO3, N2O5 and NO2, Atmospheric Measurement Techniques, 4, 1759-1776, 10.5194/amt-4-1759-2011, 2011.

McDuffie, E. E., Fibiger, D. L., Dubé, W. P., Lopez-Hilfiker, F., Lee, B. H., Thornton, J. A., Shah, V., Jaeglé, L., Guo, H., Weber, R. J., Michael Reeves, J., Weinheimer, A. J., Schroder, J. C., Campuzano-Jost, P., Jimenez, J. L., Dibb, J. E., Veres, P., Ebben, C., Sparks, T. L., Wooldridge, P. J., Cohen, R. C., Hornbrook, R. S., Apel, E. C., Campos, T., Hall, S. R., Ullmann, K., and Brown, S. S.: Heterogeneous $N_2O_5$ Uptake During Winter: Aircraft Measurements During the 2015 WINTER Campaign and Critical Evaluation of Current Parameterizations, J. Geophys. Res.: Atmos., 123, 4345-4372, 10.1002/2018jd028336, 2018.

Morgan, W. T., Ouyang, B., Allan, J. D., Aruffo, E., Di Carlo, P., Kennedy, O. J., Lowe, D., Flynn, M. J., Rosenberg, P. D., Williams, P. I., Jones, R., McFiggans, G. B., and Coe, H.: Influence of aerosol chemical composition on N2O5 uptake: airborne regional measurements in northwestern Europe, Atmos. Chem. Phys., 15, 973-990, 10.5194/acp-15-973-2015, 2015.

Tham, Y. J., Wang, Z., Li, Q. Y., Wang, W. H., Wang, X. F., Lu, K. D., Ma, N., Yan, C., Kecorius, S., Wiedensohler, A., Zhang, Y. H., and Wang, T.: Heterogeneous N2O5 uptake coefficient and production yield of ClNO2 in polluted northern China: roles of aerosol water content and chemical composition, Atmos. Chem. Phys., 18, 13155-13171, 10.5194/acp-18-13155-2018, 2018.

Wang, H., Chen, J., and Lu, K.: Development of a portable cavity-enhanced absorption spectrometer for the measurement of ambient NO3 and N2O5: experimental setup, lab characterizations, and field applications in a polluted urban environment, Atmospheric Measurement Techniques, 10, 1465-1479, 10.5194/amt-10-1465-2017, 2017a.

Wang, H., Lu, K., Chen, X., Zhu, Q., Chen, Q., Guo, S., Jiang, M., Li, X., Shang, D., Tan, Z., Wu, Y., Wu, Z., Zou, Q., Zheng, Y., Zeng, L., Zhu, T., Hu, M., and Zhang, Y.: High $N_2O_5$ Concentrations

Observed in Urban Beijing: Implications of a Large Nitrate Formation Pathway, Environ Sci Tech Let, 4, 416-420, 10.1021/acs.estlett.7b00341, 2017b.

Wang, H., Chen, X., Lu, K., Tan, Z., Ma, X., Wu, Z., Li, X., Liu, Y., Shang, D., Wu, Y., Zeng, L., Hu, M., Schmitt, S., Kiendler-Scharr, A., Wahner, A., and Zhang, Y.: Wintertime N2O5 uptake coefficients over the North China Plain, Science Bulletin, 65, 765-774, https://doi.org/10.1016/j.scib.2020.02.006, 2020.

Wang, H. C., Lu, K. D., Guo, S., Wu, Z. J., Shang, D. J., Tan, Z. F., Wang, Y. J., Le Breton, M., Lou, S. R., Tang, M. J., Wu, Y. S., Zhu, W. F., Zheng, J., Zeng, L. M., Hallquist, M., Hu, M., and Zhang, Y. H.: Efficient N2O5 uptake and NO3 oxidation in the outflow of urban Beijing, Atmos. Chem. Phys., 18, 9705-9721, 10.5194/acp-18-9705-2018, 2018a.

Wang, W., Wang, Z., Yu, C., Xia, M., Peng, X., Zhou, Y., Yue, D., Ou, Y., and Wang, T.: An in situ flow tube system for direct measurement of N2O5 heterogeneous uptake coefficients in polluted environments, Atmospheric Measurement Techniques, 11, 5643-5655, 10.5194/amt-11-5643-2018, 2018b.

Wang, X., Wang, H., Xue, L., Wang, T., Wang, L., Gu, R., Wang, W., Tham, Y. J., Wang, Z., Yang, L., Chen, J., and Wang, W.: Observations of $N_2O_5$ and $ClNO_2$ at a polluted urban surface site in North China: High $N_2O_5$ uptake coefficients and low $ClNO_2$ product yields, Atmos. Environ., 156, 125-134, 10.1016/j.atmosenv.2017.02.035, 2017c.

Wang, Z., Wang, W., Tham, Y. J., Li, Q., Wang, H., Wen, L., Wang, X., and Wang, T.: Fast heterogeneous N2O5 uptake and ClNO2 production in power plant and industrial plumes observed in the nocturnal residual layer over the North China Plain, Atmos. Chem. Phys., 17, 12361-12378, 10.5194/acp-17-12361-2017, 2017d.

Xia, M., Wang, W., Wang, Z., Gao, J., Li, H., Liang, Y., Yu, C., Zhang, Y., Wang, P., Zhang, Y., Bi, F., Cheng, X., and Tao, W.: Heterogeneous Uptake of N2O5 in Sand Dust and Urban Aerosols Observed during the Dry Season in Beijing, Atmosphere, 10, 204, 10.3390/atmos10040204, 2019.

Yu, C., Wang, Z., Xia, M., Fu, X., Wang, W., Yee Jun, T., Chen, T., Zheng, P., Li, H., Shan, Y., Wang, X., Xue, L., Zhou, Y., Yue, D., Ou, Y., Gao, J., Lu, K., Brown, S., Zhang, Y., and Tao, W.: Heterogeneous $N_2O_5$ reactions on atmospheric aerosols at four Chinese sites: improving model representation of uptake parameters, Atmos. Chem. Phys., 20, 4367-4378, 10.5194/acp-20-4367-2020, 2020.

---

## Author Comment (AC2)

**Reply on RC2**

We appreciate the reviewer for the careful reading and their constructive comments on our manuscript. **As detailed below, the reviewer's comments are** normal font, **our response to the comments are shown as** *italicized font*. New or modified text is in blue.

All the line numbers refer to original version of Manuscript ID: **amt-2022-167**.

Wang et al describe an aerosol flow reactor for the measurement of the reactive uptake of $N_2O_5$ to ambient aerosol particles. This approach has already been reported in the literature and the approach taken by the authors is very similar to that previously reported. I suggest that the authors focus the paper on the specific aspects of the flow reactor system that are new and less on the aspects that are replication of prior work. In 2009, Bertram et al reported on the development of a flow reactor for measurement of the reactivity of ambient aerosol that is strikingly similar to this. In 2018, Wang et al reported on the use of an iterative box-model coupled to the flow reactor to improve the retrieval of the reactive uptake coefficients for $N_2O_5$ to ambient aerosol, which again is very similar to that used here. It is not clear what is new with this approach that would warrant a new publication. The authors need to make the case for what technological advancement has been made. It is also not clear that the uncertainty associated with the measurements have been reduced.

The authors do note that "simultaneous $N_2O_5$ measurement at both end of the flow tube" is a unique feature of this reactor. I find this statement to be misleading: 1) The measurement is NOT simultaneous. In this technique the top and the bottom of the flow tube are sampled sequentially within one duty cycle. 2) Sampling of the $N_2O_5$ concentration at the top and the bottom of the flow tube was also done in Bertram et al to retrieve daily wall loss terms (see section 3.2 of Bertram et al). The authors would need to argue that measuring the wall loss more frequently leads to a reduced uncertainty in the retrieved uptake coefficients if this is the primary technical advance of the paper.

*We thank this comment. The reviewer emphasized the similarity of our work to the two previous works and questioned the novelty of this study. While our work builds on the previous two studies and attempts to address some issues that were not well addressed in the above work.*

*First, accurate quantification of wall loss is essential for the quantification of $N_2O_5$ uptake coefficients as suggested in Bertram's paper, a better way to reduce the wall loss interference is to measure it frequently (Bertram et al., 2009, see section 4.3). This suggestion was subsequently adopted by Wang et al. but the concentration of the $N_2O_5$ source they used in determining the wall loss and the total $N_2O_5$ loss was an assumed stable value rather than an observed one. Here, we dynamically determine the $N_2O_5$ source concentration frequently, which is helpful to provide accurate data*

on quantifying the wall loss as well as the total $N_2O_5$ loss in the flow tube in each measurement cycle. Figure R1 shows the example of difference of measured $N_2O_5$ wall loss at lab condition and ambient condition.

[Figure]

*Figure R1. The derived dependence of $N_2O_5$ wall loss on RH at laboratory condition (red dots) and field measurement (blue square)*

*Second, the use of an iterative box model approach to correct the potential bias, due to side reactions in the flow tube, is the highlight of the work of Wang et al. The iterative box model calculates backward to estimate the concentration of $NO_2$ and $O3$ before entering the flow tube,* **with estimated NO profile and measured $N_2O_5$ at the exit of flow tube.** *Here, the observed concentrations of $NO_2$, $O3$, and $NO$ at "time zero" are obtained through programmed cyclic measurements, which can reduce the uncertainties by adding the model constrain (more details can be found in the reply for the following Question NO. 4). Figure R2(a) presents the box whisker of $N_2O_5$ and NO concentration before the entrance during a field campaign. The variation of initial $N_2O_5$ is much larger than that in lab condition with a very small standard deviation for $N_2O_5$ concentration (<1%, mentioned in line 180), highlights the influence of ambient air mass (e.g. varying NO and temperature) to injected $N_2O_5$. Figure 2R(b) shows the case of large underestimation by using a fixed initial $N_2O_5$ concentration rather than measured values as box model input. In addition, we simulate $NO3$-$N_2O_5$ relationship via specific reactions rather than approximating it in equilibrium and introducing the equilibrium coefficient (Keq) into calculation. Determining $NO3$ or $N_2O_5$ concentration by Keq could induce large bias (up to 90%) under the high aerosol loading and low temperature (Chen et al., 2021).*

[Figure]

*Figure R2. (a) the box whisker of $N_2O_5$ source and NO measured before the entrance; (b) the inter-comparison of derived $N_2O_5$ uptake coefficient by using a fixed $N_2O_5$ and a dynamic measured $N_2O_5$ at the time zero in the iterative box model.*

*Third, to achieve the programmed cyclic measurement of key parameters such as $N_2O_5$, NOx and O3, we adopted a new design scheme of Y-tee and cyclic measurement setup, which were never used in the previous two studies.*

*In summary, we believe that the above innovation points of our study are advanced, while we may not address our innovation points well in the previous version. Thus, we have added an explanation of the novelty in the revised manuscript as follows.*

"The design of sampling module and aerosol flow tube in this work follows previous work for measuring $\gamma(N_2O_5)$ on ambient aerosols (e.g. Bertram et al., 2009). The major distinctions of this system from previous work are continuous monitor of NOx and O3 concentration before the inlet of flow tube (after sampling air mixing with $N_2O_5$ source) and the sequential measurements of $N_2O_5$ concentration both at the inlet and the exit of flow tube within a duty cycle. Constraints of these variables during the subsequent data processing can improve the measuring accuracy."

"Overall, the introduction of box model method in this study is able to effectively avoid the underestimation caused by the lack of consideration of side reactions in the flow tube. Although an iterative box model, including backward and forward simulation, has been applied to the $\gamma(N_2O_5)$ measurements via an aerosol flow tube in polluted environments (Wang et al., 2018), the box model method combined with current flow tube system in this study can improve measuring accuracy on some aspects. First, we simulate $NO_3$-$N_2O_5$ relationship via specific reactions rather than approximating it in equilibrium and introducing the equilibrium coefficient ($K$eq) into calculation. Determining $NO_3$ or $N_2O_5$ concentration by $K$eq could induce large bias (up to 90%) under the high aerosol loading and low temperature (Chen et al., 2021). Second, it is more accurate to constrain the box model with directly measured NOx, O3 and $N_2O_5$ at the entrance of the flow tube. Under a real atmosphere, the initial $N_2O_5$ concentration

after mixing with sampling air is not expected to be as stable as that in laboratory tests, due to the variations in temperature, NO concentration and other related parameters. Numerical simulations based on a constant initial $N_2O_5$ through backward simulation could then lead to significant uncertainty in $\gamma(N_2O_5)$ retrieval without careful data filtering."

*For the misleading statement pointed out by the reviewer, we replaced the word "simultaneous" with "sequential" thoroughly in the main text, and rephrased related statements.*

*The purpose of $N_2O_5$ measurement at both the top and bottom of the flow tube was not only for frequent wall loss determination, but reducing the uncertainty from $N_2O_5$ variation at the top of the flow tube due to air mass changing. During the field campaign, the ambient RH could vary from 21% to 40% within one day, which would produce around $5 \times 10^{-4}$ $s^{-1}$ difference on wall loss rate constant, leading to 50% bias for the scenario $\gamma(N_2O_5)$ of 0.02 and Sa of 800 $\mu m^2 \cdot cm^{-3}$. Similarly, we also found that the $N_2O_5$ at the inlet of the flow tube frequently varied over 0.5 ppbv due to strong NO emission, which could bias the $\gamma(N_2O_5)$ retrieval for an extra 20%. Therefore, the uncertainty of $\gamma(N_2O_5)$ measurement would be much higher without the $N_2O_5$ measurement at both the top and bottom of the flow tube than what we determined in this work (see also in section 5).*

There are a few aspects of the reported work and new directions that the authors could take this work that are (or would be) interesting:

- The residence time modeling in the flow tube was interesting, especially the conclusion that there are two flow paths. I think there is room for advancement in this technique if the distribution of reaction times was narrowed, while still preserving a long interaction time. Alternatively, it would also be interesting to try to use the RTD that is modeled within the framework of the $N_2O_5$ retrieval as it is not clear to me that an average residence time is appropriate with this type of RTD.

*Thanks for your suggestion. The RTD determined in this work is similar with that in previous flow tube system (Lambe et al., 2011;Wang et al., 2018), in which the average residence time was also used for calculating reaction rate parameters. We admit that such a simplification could lead to bias in the calculation. Therefore, we retrieved $\gamma(N_2O_5)$ by using RTD in the framework of this work and the result shows that the use of mean residence time produces 32% underestimation of $\gamma(N_2O_5)$ in the basic scenario (see table 3 in the main text). The uncertainty analysis on the use of mean residence time is presented in section 5 and as follows.*

"In addition, the mean residence time used in the box model method could bias the retrieved $\gamma(N_2O_5)$ due to the non-normal distribution of residence time with a

discernable tail. The reactants entrained by those slower streamlines close to the wall will take much longer time to reach the exit of the flow tube than that by the centerline. To evaluate the uncertainty caused by the distribution of residence time, we first performed simulations of $N_2O_5$ decay in the flow tube under the basic scenarios and calculate the exit $N_2O_5$ concentration according to the probability distribution function derived from RTD profile. Then the $\gamma(N_2O_5)$ can be retrieved from the box model method running for the duration of mean residence time, constrained by this calculated exit $N_2O_5$ concentration. The result shows that the use of mean residence time produces 32% underestimation of $\gamma(N_2O_5)$ in the basic scenario. The extent of underestimation is most sensitive to the level of $\gamma(N_2O_5)$ and RH."

- While there were some nice calculations of the uncertainty in the retrieved $N_2O_5$ uptake coefficient, actual measurements are most important. I would like to see systematic evaluation of the approach in the laboratory. Some example experiments that would be extremely informative might include: i) measurement of $g(N_2O_5)$ as a function of surface area for a model compound at constant RH and NO. ii) Modulation of NO (and RH above the deliquescent point) at the inlet while flowing a constant surface area concentration of a known aerosol composition. These experiments would confirm whether the modeled uncertainty holds for experimental conditions.

*We have conducted a series of lab experiments of $\gamma(N_2O_5)$ measurement on $(NH_4)_2SO_4$ aerosols with the modulation of Sa, NO and RH levels. In the base scenario, the RH, NO and Sa were set to 50%, 0 ppbv and 600 $\mu m^2$ $cm^{-3}$, respectively, at room temperature of 295 K with $N_2O_5$ concentration of 4.0 ppbv at the entrance of flow tube. The $\gamma(N_2O_5)$ was determined to be 0.01 ±0.002. In the sensitivity experiments, the RH was modulated from 10 to 55%, NO from 0 to 6 ppbv and Sa from 400 to 1000 $\mu m^2$ $cm^{-3}$. The results show that $\gamma(N_2O_5)$ holds within the range of Sa and NO variation, and increase with RH which is consistent with previous reported values (see the Figure R4).*

[Figure]

Figure R4. (a) The dependence of $\gamma(N_2O_5)$ on RH for laboratory-generated $(NH_4)_2SO_4$ aerosols; (b) $\gamma(N_2O_5)$ measurements on lab-generated $(NH_4)_2SO_4$ aerosols under different gradients of NO.

The Sa concentration varies from 400 to 1000 $\mu m^2$ $cm^{-3}$ at each level of NO. The red points with standard deviations represent the measured values. Previously reported values are indicated in blue marks.

**References**:

Chen, X., Wang, H., and Lu, K.: Interpretation of NO 3–N 2 O 5 observation via steady state in high aerosol air mass: The impact of equilibrium coefficient in ambient conditions, Atmospheric Chemistry and Physics Discussions, 1-14, 2021.

Lambe, A., Ahern, A., Williams, L., Slowik, J., Wong, J., Abbatt, J., Brune, W., Ng, N., Wright, J., and Croasdale, D.: Characterization of aerosol photooxidation flow reactors: heterogeneous oxidation, secondary organic aerosol formation and cloud condensation nuclei activity measurements, Atmospheric Measurement Techniques, 4, 445-461, 2011.

Wang, W., Wang, Z., Yu, C., Xia, M., Peng, X., Zhou, Y., Yue, D., Ou, Y., and Wang, T.: An in situ flow tube system for direct measurement of N2O5 heterogeneous uptake coefficients in polluted environments, Atmospheric Measurement Techniques, 11, 5643-5655, 10.5194/amt-11-5643-2018, 2018.

---

## Author Response (AR2)

**Reply on referee #4**

We appreciate the reviewer for the careful reading and their constructive comments on our manuscript. **As detailed below, the reviewer's comments are** normal font, **our response to the comments are shown as** *italicized font*. **New or modified text is in blue.**

All the line numbers refer to original version of Manuscript ID: **amt-2022-167**.

It is critical to developing in-situ $\gamma$(N2O5) measuring system with high accuracy since direct measurement of $\gamma$(N2O5) on ambient aerosols is still very scarce. Chen et al describe an aerosol flow reactor system combined with a box model to determine $\gamma$(N2O5) on ambient aerosols. As far as I know, this is the third direct measurement system of ambient $\gamma$(N2O5) which was successfully applied to field measurement. The system was built on two previous works (Bertram et al, AMT, 2009 and Wang et al., AMT, 2019) and has some similarities. The authors have made efforts to reduce the measurement uncertainty and improve the performance of the instrument, such as using periodical measurements of N2O5 source concentration at the entrance of the flow tube by using a well-designed tubing connection. N2O5 is very sensitive to NO, relative humidity, and temperature, which biases the entrance N2O5 concentration even within a short time period, and thus the $\gamma$(N2O5) results. The authors ran a detailed box model with the constraint of NO, NO2 and O3 at 'time zero' to retrieve N2O5 loss rate. These improvements can largely reduce the uncertainty of $\gamma$(N2O5) retrieval from varying ambient air mass and expand its adaptability and application scope. This work is a small step forward in the direct measurement of N2O5 uptake and can really contribute to the community. Overall, this paper is well written with detailed and comprehensive lab characterization and shows the feasibility of the performance in field applications. I recommend this manuscript published after attention to the following comment and minor mistakes.

1. To underscore the reduced uncertainty made by this instrument and further improve the readability, it is suggested that the authors refine the results of laboratory characterizations in sections 2 & 4 and compare the uncertainty and detection limit of this approach, if it is possible, with previous works.

*Thanks for the valuable suggestions. We try our best to refine the description of flow tube system and its laboratory characterization tests in sections 2 & 4 according to reviewer's suggestion. There is no reported $\gamma(N_2O_5)$ detection limit in previous ambient flow tube studies. Thus, we only add the comparison of uncertainty with previous studies in section 5 as follows.*

"To directly compare with previous studies, at 0.03 $\gamma(N_2O_5)$ with 1000 $\mu m^2$ $cm^{-3}$ Sa, the uncertainty is calculated to be 19% which is lower than that ~24% in Bertram et al (2009) and that ranging 37%~40% in Wang et al (2018)."

2. Line 138: delete 'to'.

*We revise the sentence according to the reviewer's suggestion.*

3. Line 144: What is the advantage of using a static mixer? Please clarify its effect.

*We clarify the advantages and effects of using a static mixer in front of the entrance as follows.*

"A 10 cm long stainless-steel static mixer is mounted inside the Y-tee in order to swirl the flow and therefore facilitate the mixing between sampling stream and $N_2O_5$ source in a relatively short distance. The presence of static mixer ahead the inlet also help to improve the flow expansion after entering the flow tube by minimizing recirculation zone, which decreases the wall loss of $N_2O_5$ and particles (Huang et al., 2017)."

4. Line 301-302: Change to 'be fully drained out of the flow tube'.

*We revise the sentence according to the reviewer's suggestion.*

5. Line 367-369: Please clarify the temperature, RH, and the range of NO2 and O3 used in these tests.

*We thank this suggestion and clarify the conditions of these tests according to the reviewer's suggestion as follows.*

"The levels of $PNO_3$ was adjusted by $NO_2$ and $O_3$ concentrations ($O_3$ ranging from 10 to 80 ppbv and $NO_2$ ranging from 50 to 160 ppbv) under the temperature of 283 K and RH of 30%."